# NEURAL SUM-OF-SQUARES: CERTIFYING THE NONNEGATIVITY OF POLYNOMIALS WITH TRANSFORMERS

**Nico Pelleriti**[1,2,*]   **Christoph Spiegel**[1,2]   **Shiwei Liu**[3,4,5]   **David Martínez-Rubio**[6]
**Max Zimmer**[1,2]   **Sebastian Pokutta**[1,2]

[1] Zuse Institute Berlin   [2] Technical University of Berlin   [3] ELLIS Institute Tübingen
[4] Max Planck Institute for Intelligent Systems   [5] Tübingen AI Center
[6] Universidad Carlos III de Madrid

## ABSTRACT

Certifying nonnegativity of polynomials is a well-known NP-hard problem with direct applications spanning non-convex optimization, control, robotics, and beyond. A sufficient condition for nonnegativity is the Sum of Squares (SOS) property, i.e., it can be written as a sum of squares of other polynomials. In practice, however, certifying the SOS criterion remains computationally expensive and often involves solving a Semidefinite Program (SDP), whose dimensionality grows quadratically in the size of the monomial basis of the SOS expression; hence, various methods to reduce the size of the monomial basis have been proposed. In this work, we introduce the first learning-augmented algorithm to certify the SOS criterion. To this end, we train a Transformer model that predicts an almost-minimal monomial basis for a given polynomial, thereby drastically reducing the size of the corresponding SDP. Our overall methodology comprises three key components: efficient training dataset generation of over 100 million SOS polynomials, design and training of the corresponding Transformer architecture, and a systematic fallback mechanism to ensure correct termination, which we analyze theoretically. We validate our approach on over 200 benchmark datasets, achieving speedups of over $100\times$ compared to state-of-the-art solvers and enabling the solution of instances where competing approaches fail. Our findings provide novel insights towards transforming the practical scalability of SOS programming. Code is available at `https://github.com/ZIB-IOL/Neural-Sum-of-Squares`.

## 1 INTRODUCTION

Global optimization of high-degree, *nonconvex* polynomials underpins tasks as diverse as satellite-attitude control (Misra et al., 2020; Gollu, 2008; Jarvis-Wloszek et al., 2003), energy-shaping of quadrotors, and control theory (Bramburger et al., 2024; Jarvis-Wloszek et al., 2003). Unconstrained polynomial optimization reduces to certifying nonnegativity: the value of $\min_{\mathbf{x} \in \mathbb{R}^n} p(\mathbf{x})$ is the largest $\gamma$ such that $p(\mathbf{x}) - \gamma \geq 0$ for all real $\mathbf{x}$. To illustrate this connection, consider the simple polynomial

$$p(x_1, x_2) = 4x_1^4 + 12x_1^2 x_2^2 + 9x_2^4 + 1.$$

We can verify that $p(x_1, x_2) - \gamma$ is nonnegative for all $\gamma \leq 1$ by rewriting it as $(2x_1^2 + 3x_2^2)^2 + 1 - \gamma \geq 1 - \gamma$. Since this expression is positive when $\gamma < 1$ and zero when $\gamma = 1$, we conclude that $\gamma = 1$ is the global minimum of the polynomial. While this example admits a straightforward algebraic verification, deciding nonnegativity is *NP-hard* in general, even for simple quartic polynomials such as the one above, which motivates the use of convex relaxations (Ahmadi et al., 2011).

A widely used convex relaxation for nonnegativity certification is the SOS condition: a polynomial $p(\mathbf{x})$ is a SOS if it can be written as $p(\mathbf{x}) = \sum_{i=1}^{r} h_i(\mathbf{x})^2$ for some polynomials $h_i(\mathbf{x})$. Since any SOS polynomial is clearly nonnegative, this provides a sufficient condition for nonnegativity. For our example, we used that $p(x_1, x_2) = (2x_1^2 + 3x_2^2)^2 + 1^2$, which is SOS and therefore nonnegative.

---

[*]Correspondence to `pelleriti@zib.de`.

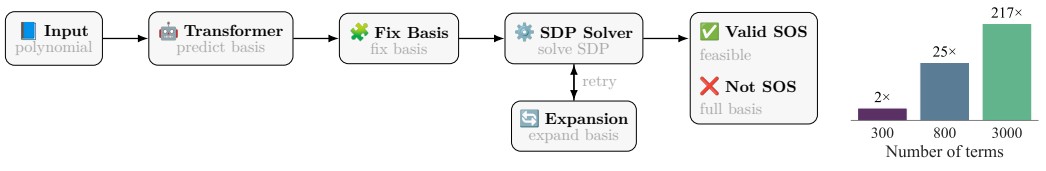

(i) Schematic of the learning-augmented SOS framework.  (ii) Speedup.

Figure 1: Overview of our approach for SOS verification. (i) Pipeline schematic: given a polynomial, a Transformer predicts a compact basis, then the basis is adjusted to ensure necessary conditions are met, and an SDP is solved with iterative expansion if needed. The method guarantees correctness: if a SOS certificate exists, it will be found; otherwise, infeasibility is certified at the full basis. (ii) Speedup over baseline for sparse polynomials with 300, 800, and 3000 distinct terms.

Parrilo (2003) proved that this condition can be checked via an SDP. That is, $p$ is SOS if and only if there exists a Positive Semidefinite (PSD) matrix $Q$ such that $p(\mathbf{x}) = \mathbf{z}(\mathbf{x})^\top Q \mathbf{z}(\mathbf{x})$, where $\mathbf{z}(\mathbf{x})$ is a vector of monomials (polynomial terms like $x_1^2$, $x_1 x_2$, etc.). Crucially, SDPs can be solved in polynomial time in the dimension of $\mathbf{z}(\mathbf{x})$, making the choice of monomial vector central to computational efficiency.

For our running example $p(x_1, x_2) = 4x_1^4 + 12x_1^2 x_2^2 + 9x_2^4 + 1$, the standard monomial vector contains all monomials up to half the degree of the polynomial, yielding $\mathbf{z}(\mathbf{x}) = [1, x_1, x_2, x_1 x_2, x_1^2, x_2^2]^\top$. However, we can compute the same SOS decomposition using the much smaller monomial vector $\mathbf{z}'(\mathbf{x}) = [1, x_1^2, x_2^2]^\top$. This gives us $p(\mathbf{x}) = \mathbf{z}'(\mathbf{x})^\top Q \mathbf{z}'(\mathbf{x})$ with the matrix

$$Q = \begin{pmatrix} 1 & 0 & 0 \\ 0 & 4 & 6 \\ 0 & 6 & 9 \end{pmatrix}.$$

Therefore, identifying a compact monomial basis $\mathbf{z}'(\mathbf{x})$ enables a drastic reduction in the size of the resulting SDP, yielding substantial computational savings.

Unfortunately, finding such compact bases is itself challenging, and a long line of work has focused on reducing basis size (Reznick, 1978; Waki et al., 2006; Lofberg, 2009; Wang et al., 2021b). In our example, the commonly used *Newton polytope* with diagonal consistency (Lofberg, 2009) method (a geometric approach that identifies relevant monomials based on the polynomial's structure) would yield the basis $\mathbf{z}_N(\mathbf{x}) = [1, x_1^2, x_2^2, x_1 x_2]^\top$, which is significantly smaller than the full basis $\mathbf{z}(\mathbf{x})$ but still not *minimal*.[1]

In this paper, we address the problem of efficiently selecting compact monomial bases by introducing a learning-augmented approach to SOS basis selection. We pose basis selection as a prediction task: given a polynomial, predict a compact monomial basis that is sufficient to express the polynomial as an SOS and yields an SDP that is as small as possible. Figure 1 provides an overview of the proposed approach.

To this end, we train a Transformer model (Vaswani et al., 2017) that takes a tokenized polynomial $p$ as input and outputs a monomial basis $B$. The training data consists of millions of SOS polynomials and (near-)minimal basis pairs, which we generate efficiently through a reverse sampling process that allows us to construct a polynomial $p$ with a known compact SOS decomposition and monomial basis.

While our learning-based approach offers the potential for finding significantly smaller bases than rule-based methods, machine learning predictions are inherently inexact. When our Transformer model predicts an incomplete basis—one missing essential monomials for a valid SOS decomposition—we employ a systematic repair strategy that preserves correctness guarantees. Specifically, if the initially predicted basis $B$ cannot produce a valid SOS decomposition, we iteratively expand $B$ by adding monomials until either (1) we discover a valid decomposition, or (2) all candidate monomials have been considered, implying no SOS decomposition exists. This fallback mechanism

---

[1]A basis is minimal if it is the smallest possible basis for which a given polynomial $p$ admits an SOS decomposition. To the best of our knowledge, the computational complexity of finding the true minimal basis remains open, though we derive tight lower bounds on minimal basis size.

ensures that our method maintains the same correctness guarantees as traditional approaches: it correctly identifies whether a SOS decomposition exists, while achieving significant speedups through accurate predictions. We summarize our contributions as follows.

**Learning-augmented SOS programming.**  We introduce a novel learning-augmented algorithm that addresses the fundamental computational bottleneck in SOS programming. Our approach uses a Transformer model to predict compact monomial bases for constructing SOS decompositions, directly reducing the size of the resulting SDP problems. By identifying sparse monomial structures that traditional rule-based methods miss, our approach achieves significant computational speedups while maintaining correctness guarantees.

**Theoretical guarantees.**  We provide a theoretical analysis, showing that the algorithm's worst-case computational cost exceeds the standard baseline by at most a constant factor even when the machine learning predictions are completely incorrect. Further, we show that the computational efficiency of the algorithm is directly related to the quality of the predicted basis.

**Empirical evaluation.**  We validate our approach across different polynomial structures on over 200 benchmark datasets, including problems with up to 100 variables, training on over 100 million polynomials. Our experiments demonstrate multiple orders of magnitude speedups compared to state-of-the-art baselines, while being robust to distribution shifts.

## 1.1  RELATED WORK

**Sum of Squares (SOS) Optimization**  SOS programming has its theoretical foundations in seminal works by Parrilo (2003) and Lasserre (2001), who independently developed frameworks for converting polynomial optimization problems into SDPs. Subsequent work has focused on scaling these relaxations to higher dimensions and degrees. Upstream basis-reduction methods—most notably Newton-polytope pruning (Reznick, 1978) and diagonal-inconsistency checks (Lofberg, 2009)—remove unnecessary monomials before forming the SDP. Related *SOS basis pursuit* methods iteratively refine certificates via LP/SOCP-style subproblems while updating a change of basis for the Gram representation (at fixed lifted dimension) (Ahmadi & Hall, 2016; Miller et al., 2022). Our method also acts upstream, but uses a learned predictor with repair and verification to select a compact *monomial set* and thus shrink the SDP, yielding speedups with correctness guarantees. In contrast, chordal/term sparsity and structured-subset decompositions (Waki et al., 2006; Mason & Papachristodoulou, 2014; Zheng et al., 2019; Wang et al., 2021b;a; Newton & Papachristodoulou, 2022) and accelerations such as AnySOS (Driggs & Fawzi, 2019) primarily improve the efficiency of solving a given SOS formulation, especially in control settings (Dai & Sznaier, 2021; Strasser et al., 2021). More recently, Li et al. (2025) study the use of Large Language Models (LLMs) for SOS-related mathematical reasoning. They introduce a benchmark of approximately 1,000 polynomials together with expert-designed reasoning prompts, and show that structured prompting and supervised fine-tuning can substantially improve direct SOS classification accuracy. In contrast to their setting, our goal is not to have an LLM directly decide whether a polynomial is SOS from the polynomial itself using prompting and procedural algebraic checks. Instead, we use learning upstream inside a verified optimization pipeline: given a polynomial, our model predicts a compact monomial basis, after which we solve an SDP and use repair to retain correctness guarantees.

**Machine Learning and *Algorithms with Predictions* for Combinatorial Optimization**  Recent advances in deep learning have demonstrated effectiveness in addressing combinatorial optimization problems, with notable successes in routing (Kool et al., 2018) and mixed integer programming (Nair et al., 2020). To our knowledge, our approach represents the first application of learning-augmented algorithms to SOS programming, where the fundamental computational bottleneck lies in selecting compact monomial bases from exponentially large candidate sets. We analyze our approach within the *Algorithms with Predictions* framework (Roughgarden, 2021), which provides theoretical foundations for quantifying the performance of learning-augmented algorithms that integrate potentially inexact predictions while maintaining worst-case guarantees. We build upon neural approaches for computational algebra developed by Kera et al. (2025b), specifically adapting their polynomial tokenization and sequence generation techniques for our monomial basis prediction task. More broadly, our work is situated in an emerging line of research at the intersection of computa-

tional algebra, symbolic computation, and machine learning (Kera et al., 2024; 2025a; Pelleriti et al., 2025; Alfarano et al., 2024).

## 2 NOTATION AND PRELIMINARIES

Vectors of real numbers and variables are denoted in bold, and indexed by subscripts. We denote the set of all monomials of degree $d \in \mathbb{N}$ or less by $\mathcal{M}_d$. For a subset $B \subseteq \mathcal{M}_d$ we denote the vector of the monomials in $B$ by $\mathbf{z}_B(\mathbf{x}) \in \mathbb{R}^{|B|}$. Further, for a given polynomial of degree $d$ with $n$ variables $p \in \mathbb{R}[x_1, \ldots, x_n]$, we denote its monomial support, i.e., all monomials appearing in $p$, by $S(p) \subseteq \mathcal{M}_d$. A matrix symmetric $Q \in \mathbb{R}^{k \times k}$ is PSD if $\mathbf{z}^\top Q \mathbf{z} \geq 0$ for all $\mathbf{z} \in \mathbb{R}^k$, denoted $Q \succeq 0$.

For a SOS polynomial $p(\mathbf{x}) = \mathbf{z}_B(\mathbf{x})^\top Q \mathbf{z}_B(\mathbf{x})$ with $Q \succeq 0$, we call $B$ the *basis* and $\mathbf{z}_B(\mathbf{x})$ the *monomial basis vector*. The Newton polytope $N(p)$ is the convex hull of exponent vectors of monomials in $p$. For our running example $p(x_1, x_2) = 4x_1^4 + 12x_1^2 x_2^2 + 9x_2^4 + 1$, the Newton polytope $N(p)$ is the convex hull of points $\{(4,0), (2,2), (0,4), (0,0)\}$ corresponding to the monomials $x_1^4, x_1^2 x_2^2, x_2^4, 1$. For any SOS decomposition $p(\mathbf{x}) = \mathbf{z}_B(\mathbf{x})^\top Q \mathbf{z}_B(\mathbf{x})$, the exponents of basis monomials in $B$ must lie within the half Newton polytope $\frac{1}{2}N(p)$ (Reznick, 1978), i.e., $N(p)$ scaled by a factor of $1/2$ about the origin. Consequently, a valid basis can be chosen from the finite set $\left\{ \mathbf{x}^\alpha : \alpha \in \left( \frac{1}{2}N(p) \right) \cap \mathbb{Z}_{\geq 0}^n \right\}$, though many such monomials are typically redundant.

## 3 METHODOLOGY

We present a learning-augmented algorithm for SOS basis selection. This is complementary to recent LLM-based approaches such as Li et al. (2025), which study direct SOS reasoning from the polynomial itself. Our method uses a Transformer to predict a compact monomial basis, which is then verified through SDP solving. However, these predictions can be incorrect. When the predicted basis is incomplete, the resulting SDP may be infeasible even though the polynomial has a valid SOS decomposition. Therefore, we need a systematic repair mechanism to handle incorrect predictions while maintaining the computational speedups from accurate predictions. Our approach addresses this challenge in three stages.

(i) **Transformer-based basis prediction.** Given a polynomial $p$, a trained Transformer model predicts a compact set of monomials likely to form a valid SOS basis.

(ii) **Construction of a valid initial basis.** We verify that the predicted basis satisfies necessary structural conditions. If not, we employ a greedy heuristic to augment the basis until these conditions are satisfied.

(iii) **Iterative expansion with verification.** We solve the SOS problem using the current basis. If the solver fails, we expand the basis using a learnable scoring mechanism and resolve the SDP. This process iterates until either a valid SOS decomposition is found or all candidate monomials are exhausted, certifying that the polynomial is not SOS.

Figure 1 illustrates the complete workflow of our approach. In the following sections, we describe each of these three components above in detail.

### 3.1 TRANSFORMER-BASED BASIS PREDICTION

To train the Transformer, we require supervised pairs of polynomials $p$ and their SOS bases $B \subseteq \mathcal{M}_d$. Because finding *near-minimal* SOS bases is computationally hard, we generate polynomials with *known compact* bases via reverse sampling: sample a monomial set $B \subseteq \mathcal{M}_d$, draw $Q \succeq 0$, and set $p(\mathbf{x}) = \mathbf{z}_B(\mathbf{x})^\top Q \mathbf{z}_B(\mathbf{x})$. This guarantees $B$ is a valid SOS basis for $p$ by construction and often yields near-minimal bases that match our lower bounds in Appendix F (cf. Appendix E.6 for empirical verification, demonstrating that these bases are near-minimal).

To ensure comprehensive coverage, we vary the PSD structure of $Q$ across four categories: dense full-rank matrices, unstructured sparse matrices, block diagonal matrices, and low-rank matrices, as illustrated in Figure 2.

We further diversify our training data by varying the number of variables, polynomial degree, and number of terms, which yields polynomial families that span dense, sparse, and clique-structured polynomials across different problem scales. To validate the quality of our training data, we verify that the generated bases achieve sizes close to theoretical lower bounds derived in Lemma 7 and Lemma 8, confirming that these bases are near-minimal. Lemma 7 and Lemma 8 formalize structural constraints that any SOS basis must satisfy, providing lower bounds and consistency relations that guide our reverse-sampling construction. These results ensure that the generated training bases are not only valid but close to theoretically minimal, which is essential for training a model that generalizes well to compact bases. Complete details regarding parameter grids, seeds, and sample counts are provided in Appendix D.

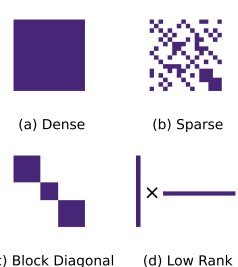

(a) Dense  (b) Sparse

(c) Block Diagonal  (d) Low Rank

Figure 2: Sampled matrix structures.

We formulate basis prediction as a sequence generation problem. Given a polynomial $p$, our Transformer model predicts a monomial basis $B$ from the half Newton polytope $\frac{1}{2}N(p)$ that likely admits an SOS decomposition $p(\mathbf{x}) = \mathbf{z}_B(\mathbf{x})^\top Q \, \mathbf{z}_B(\mathbf{x})$ for some $Q \succeq 0$, i.e.

$$\text{TRANSFORMER}(p) \mapsto B \subseteq \frac{1}{2}N(p) \cap \mathbb{Z}_{\geq 0}^n.$$

We tokenize polynomials using the scheme from Kera et al. (2025b). Each monomial is represented by its coefficient (prefixed with C) followed by the exponents of each variable (prefixed with E). For our running example $p(x_1, x_2) = 4x_1^4 + 12x_1^2x_2^2 + 9x_2^4 + 1$, the tokenized input is

C4.0 E4 E0 + C12.0 E2 E2 + C9.0 E0 E4 + C1.0 E0 E0.

The model generates a sequence of monomials separated by SEP tokens, representing the predicted basis. For our example, the target output sequence is

E2 E0 SEP E0 E2 SEP E0 E0,

which corresponds to the minimal basis $\{x_1^2, x_2^2, 1\}$. The model uses monomial-embeddings to map tokenized terms like C1.0 E4 E0 to vector representations, significantly reducing sequence lengths. We generate basis monomials autoregressively until producing an end-of-sequence token EOS.

**Remark 1.** *Our results also indicate that the method does not fundamentally depend on Transformers. In Appendix E.9 we compare LSTMs (Hochreiter & Schmidhuber, 1997) and Mamba state-space models (Gu & Dao, 2023; Dao & Gu, 2024) under a fixed 5-hour training budget, with a similar embedding scheme as for the Transformer. Both alternatives achieved reasonable performance, with Mamba in particular being competitive with the Transformer (cf. Figure 12 and Table 30). This suggests that other sequence-to-sequence architectures may also be suitable for SOS basis prediction and represent a promising direction for future work.*

### 3.2 CONSTRUCTION OF A VALID INITIAL BASIS

Before attempting to solve the SDP with the predicted basis, we first check whether it satisfies the following necessary condition that any valid SOS basis must satisfy.

**Lemma 2.** *Let $p(\mathbf{x}) \in \mathbb{R}[x_1, \ldots, x_n]$ be a polynomial and $B \subseteq \mathcal{M}_d$ be a basis. Then, the support of $p$ is contained in the set of all pairwise products of monomials in $B$, formally $S(p) \subseteq B \cdot B$ where $B \cdot B = \{m_i \cdot m_j : m_i, m_j \in B\}$. We say that $B$ covers $p(\mathbf{x})$ if $S(p) \subseteq B \cdot B$.*

We defer the proof to Appendix F. Continuing our running example, let $p(x_1, x_2) = 4x_1^4 + 12x_1^2x_2^2 + 9x_2^4 + 1$ with candidate basis $B = \{1, x_1^2, x_1x_2\}$. The support is $S(p) = \{1, x_1^4, x_2^4, x_1^2x_2^2\}$, and the pairwise products are

$$B \cdot B = \{1, x_1^2, x_1x_2, x_1^4, x_1^3x_2, x_1^2x_2^2\}.$$

Since $x_2^4 \in S(p)$ but $x_2^4 \notin B \cdot B$, the basis $B$ cannot represent $p$ as a sum of squares.

If the predicted basis does not cover all monomials in $S(p)$, we extend it using a greedy repair algorithm that adds monomials covering the most missing support terms, which we call COVERAGEREPAIR and detail in Algorithm 3 in the appendix. For our example with $B = \{1, x_1^2, x_1x_2\}$

and missing monomial $x_2^4$, we could add $x_2^2$ since $x_2^2 \cdot x_2^2 = x_2^4$, allowing us to cover the missing term. This process continues until all monomials in $S(p)$ can be expressed as pairwise products of basis elements, ensuring that $S(p) \subseteq B \cdot B$ as required by Lemma 2.

Algorithm 1 summarizes the initial construction phase. First, we obtain a predicted basis $B_0$ from the TRANSFORMER model. Since $B_0$ may not satisfy the coverage requirement from Lemma 2, we extend it using the COVERAGEREPAIR routine to ensure that all monomials in the polynomial's support $S(p)$ can be expressed as pairwise products of basis elements, yielding a basis $B_{\text{cov}}$ that covers all support monomials. We then solve the SDP to check feasibility. If feasible, we return the decomposition; otherwise, we proceed to expansion.

---

**Algorithm 1** Initial Basis Construction

**Input:** Polynomial $p$
1: $B_0 \leftarrow \text{TRANSFORMER}(p)$
2: $B_{\text{cov}} \leftarrow \text{COVERAGEREPAIR}(B_0, S(p))$
3: **if** $\text{SOLVESDP}(B_{\text{cov}}, p)$ is feasible **then**
4:     **return feasible**
5: **else**
6:     **return infeasible**
7: **end if**

---

### 3.3 ITERATIVE BASIS EXPANSION WITH VERIFICATION

If Algorithm 1 fails to find a feasible SDP solution with the coverage-repaired basis $B_{\text{cov}}$, we expand the basis by incorporating additional monomials from the candidate pool $\frac{1}{2}N(p)$. Rather than adding monomials arbitrarily, we rank each candidate monomial by a learned *score* and expand the basis in order of decreasing score.

To define the scoring function $\text{SCORE}(u, p)$ for a monomial $u$ given polynomial $p$, we exploit the fact that our Transformer is *not* equivariant with respect to variable orderings. That is, if we permute the variables in a polynomial (e.g., swap $x_1$ and $x_2$), the Transformer may predict a different basis for the mathematically equivalent problem. We use *permutation-based scoring*: run the Transformer on $L$ random variable permutations $\Pi = \{\pi_1, \ldots, \pi_L\}$ of the variables in the polynomial, then score each monomial $u$ by its frequency across predictions:

$$\text{SCORE}(u, p) = \frac{1}{L} \sum_{i=1}^{L} \mathbf{1}[u \in \pi_i^{-1}(B_{\pi_i})],$$

where $B_{\pi_i}$ is the basis predicted for the $\pi_i$-permuted polynomial and $\pi_i^{-1}(B_{\pi_i})$ is the basis obtained by undoing the permutation $\pi_i$ on $B_{\pi_i}$. Unless otherwise noted we set $L = 4$; see Appendix E.2 for an ablation showing diminishing returns beyond $L = 8$.

The score function creates a ranking of all monomials in the candidate set $\frac{1}{2}N(p)$, where higher scores indicate monomials that are more likely to belong to a valid SOS decomposition. Hence, we have to call the Transformer $L$ times on different variable permutations and aggregate the results.

Given the polynomial $p$, coverage-repaired basis $B_{\text{cov}}$ from Algorithm 1, expansion schedule $m_1 < m_2 < \cdots < m_k = |\frac{1}{2}N(p)|$, and the Newton polytope $\frac{1}{2}N(p)$ as our candidate set, we sort candidate monomials by score and create an ordered list starting with $B_{\text{cov}}$ followed by the best candidates. For each scheduled size $m_t$, we take the first $m_t$ monomials to form the basis and solve the SDP. If the SDP is feasible, we have found a valid SOS decomposition and terminate successfully; otherwise, we try the next larger size. In the worst case, we solve $k$ SDPs, with the last equivalent to the standard Newton polytope approach. Algorithm 2 summarizes this expansion process.

---

**Algorithm 2** Ordered Expansion

**Input:** Polynomial $p$; initial basis $B_{\text{cov}}$; schedule $m_1 < m_2 < \cdots < m_k$;
1: $C \leftarrow \text{sort}(\frac{1}{2}N(p) \setminus B_{\text{cov}}, \text{SCORE}(\cdot, p))$
2: $M \leftarrow \text{concat}(B_{\text{cov}}, C)$
3: **for** $t \in \{2, \ldots, k\}$ **do**
4:     $B \leftarrow M[1 : m_t]$
5:     **if** $\text{SOLVESDP}(B, p)$ feasible **then**
6:         **return feasible**
7:     **end if**
8: **end for**
9: **return infeasible**

---

# 4 EXPERIMENTS

Table 1: Average results on representative configurations. Left: basis size (monomials). Right: solve time (s). Ours is the proposed method (with repair); Newton is the Newton polytope; Full is the dense full basis. MOSEK is used, except for low-rank cases where SCS is employed due to numerical issues. "–" indicates solver failure/timeout. Speedup is ours over Newton.

| Structure | $n$ | $d$ | $m$ | $\|B^*\|$ | Average basis size | | | Average solve time (s) | | | Speedup |
|---|---|---|---|---|---|---|---|---|---|---|---|
| | | | | | **Ours** | **Newton** | **Full** | **Ours** | **Newton** | **Full** | |
| dense | 4 | 6 | 20 | 19 | **19** | 21 | 35 | 0.4 | **0.28** | 0.67 | 0.7 |
| | 6 | 12 | 30 | 30 | **33** | 48 | 924 | **1.01** | 2.44 | – | 2.42 |
| | 8 | 20 | 30 | 28 | 38 | **32** | 43,758 | **3.4** | 309.4 | – | 91.00 |
| | 6 | 20 | 60 | 58 | **89** | 236 | 26,334 | **18.3** | 1,629.6 | – | 89.05 |
| sparse | 4 | 6 | 20 | 15 | **15** | 18 | 35 | 0.23 | **0.20** | 0.83 | 0.87 |
| | 6 | 12 | 30 | 26 | **27** | 40 | 924 | **0.57** | 1.20 | – | 2.11 |
| | 8 | 20 | 30 | 26 | **27** | 28 | 43,758 | **0.62** | 15.3 | – | 24.68 |
| | 6 | 20 | 60 | 56 | **73** | 233 | 26,334 | **7.39** | 1,606.4 | – | 217.39 |
| low-rank | 4 | 6 | 20 | 19 | **19** | 22 | 35 | 0.35 | **0.29** | 0.68 | 0.83 |
| | 6 | 12 | 30 | 30 | **30** | 48 | 924 | **0.71** | 25.4 | – | 35.77 |
| | 8 | 20 | 30 | 30 | 35 | **31** | 43,758 | **1.03** | 301.2 | – | 292.43 |
| | 6 | 20 | 60 | 57 | **66** | 224 | 26,334 | **39.17** | 1,563.1 | – | 39.90 |
| block-diag. | 4 | 6 | 20 | 19 | **20** | 22 | 35 | 0.32 | **0.28** | 0.67 | 0.88 |
| | 6 | 12 | 30 | 30 | **31** | 48 | 924 | **0.64** | 2.44 | – | 3.81 |
| | 8 | 20 | 30 | 30 | **30** | 38 | 43,758 | **0.81** | 17.4 | – | 21.48 |
| | 6 | 20 | 60 | 59 | **71** | 231 | 26,334 | **7.74** | 1,331.1 | – | 172.02 |

We evaluate our approach through four experiments: (1) end-to-end performance vs. baselines, (2) isolation of Transformer prediction capabilities, (3) repair mechanism robustness, and (4) large-scale scalability beyond existing solver limits and distribution shifts.

## 4.1 EXPERIMENT 1: COMPARISON TO BASELINES

We evaluate our method against Newton polytope and full basis approaches across four polynomial structures: dense, sparse, low-rank, and block-diagonal. Our evaluation uses 100 test instances per configuration, with test instances ranging from $n \in \{4, 6, 8\}$ variables, degree $d \in \{6, 12, 20\}$, and average basis sizes $m$ of 20, 30, and 60 monomials.

We use MOSEK (ApS, 2025) for SDP solving, with SCS (O'Donoghue et al., 2016) used for low-rank cases due to numerical issues, imposing 2-hour time limits for all instances. We measure both basis size and total runtime. For our method, runtime includes Transformer inference and coverage-repair overhead; for Newton polytope, it includes polytope construction; and for all methods, it includes SDP solve time.

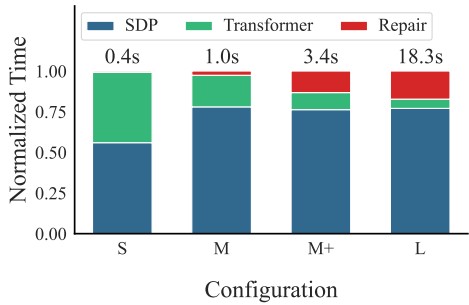

Figure 3: Time breakdown by problem size, annotated with average total solution time.

Our approach consistently achieves the smallest bases and fastest solve times, except for the smallest test cases ($n = 4$, $d = 6$, $m = 20$). Speedups range from $0.7\times$ to over $300\times$, with the full-basis method timing out on larger instances. For dense and low-rank polynomials with 8 variables and degree 20, our method maintains sub-60-second solve times, while Newton polytope requires over 1000 seconds per instance, mainly due to expensive computation of the Newton polytope (Lofberg, 2009). We also analyze the runtime breakdown in Figure 3 for the first four dense matrix configurations from Table 1, which we abbreviate as S, M, M+, L. The breakdown shows normalized times for SDP solves, Transformer inference, and repair across problem sizes. As problems become more

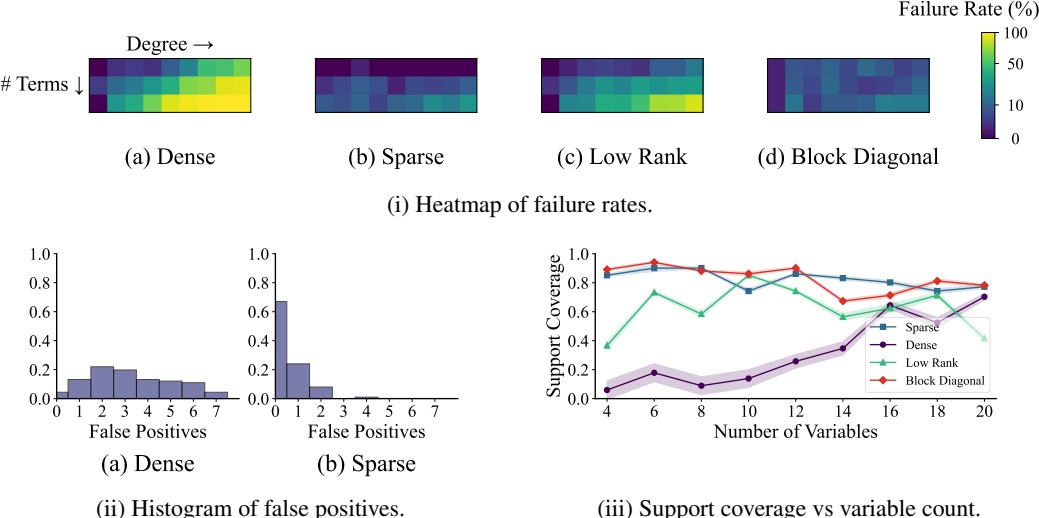

(i) Heatmap of failure rates.

(ii) Histogram of false positives.

(iii) Support coverage vs variable count.

Figure 4: Isolated Transformer performance across polynomial structures. (i) Heatmaps of failure rates over degree and structure show sparse and block-diagonal cases are easiest while dense are hardest. (ii) Histograms of false positives highlight error distributions per structure. (iii) Success probability as a function of the number of variables demonstrates monomial-embedding makes coverage consistent across variable counts.

complex, the Transformer inference takes less time relative to the total, while repair costs increase. In all but the smallest problem set, SDP solving takes up $75 - 80\%$ of the total time, except for the smallest problems (size S) where it only accounts for $55\%$. The time spent on Transformer inference and repair remains reasonable across all problem sizes. In Appendix E.12, we provide an amortization analysis of the training cost and show when the one-time learning expense is outweighed by per-instance solver speedups.

## 4.2 EXPERIMENT 2: TRANSFORMER BASIS PREDICTION

To isolate the Transformer's performance without repair, we evaluate raw basis prediction accuracy across 1000 generated instances spanning dense, sparse, low-rank, and block-diagonal structures with $n \in [4, 16]$, $d \in [4, 20]$, and average basis sizes of $10, 20$, and $30$ monomials. The Transformer's performance is highly structure-dependent (cf. Figure 4i). Sparse and block-diagonal instances achieve success rates of $85 - 95\%$, demonstrating the model's ability to exploit clear structural patterns. However, dense and low-rank matrices exhibit substantially higher failure rates (up to $100\%$ for complex instances), where cross-terms and less obvious sparsity patterns challenge the prediction mechanism. This variability underscores the importance of repair mechanisms in ensuring robust performance across diverse polynomial families. False positives are more common in dense and low-rank cases. The number of variables has minimal impact on performance, except for dense cases, which we attribute to the effectiveness of the monomial-embedding approach.

## 4.3 EXPERIMENT 3: REPAIR MECHANISM

We evaluate repair mechanisms on $n = 8$ variable, degree 20 polynomials across four repair strategies: no repair, greedy repair only, permutation-based repair (using 2 permuted versions), and combined approaches. We measure three outcomes: *exact match* (predicted basis equals ground truth), *superset* (predicted basis contains ground truth but includes extra monomials), and *insufficient* (predicted basis missing essential monomials for valid SOS decomposition).

Figure 5i shows that greedy repair achieves high exact match rates with minimal computational overhead. Permutation-based repair reduces insufficient cases but increases superset outcomes. The combined approach achieves the lowest insufficient rates across all polynomial structures, ensuring robust recovery from failed predictions.

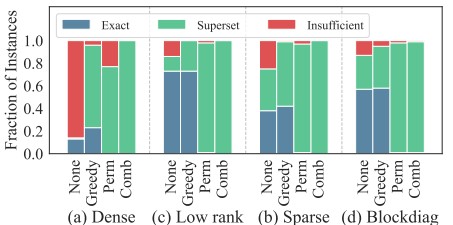
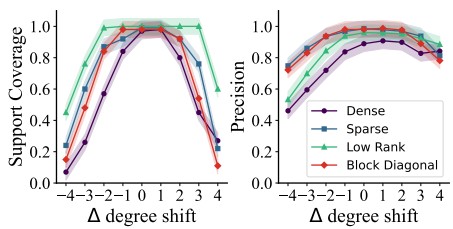

(i) Global repair outcomes by polynomial structure.

(ii) Precision metrics for repair mechanisms.

Figure 5: Repair mechanism performance evaluation. (i) Global repair outcomes demonstrate the effectiveness of greedy and permutation-based repair mechanisms across polynomial structures: permutation-based approaches often generate supersets by adding many more monomials but prove less effective for dense cases, while greedy methods show more conservative monomial addition. (ii) Distribution shift evaluation results, indicating that the combined repair mechanisms are effective.

### 4.4 EXPERIMENT 4: DISTRIBUTION SHIFT AND LARGE SCALE EVALUATION

We compare our method with state-of-the-art solvers (SumOfSquares.jl (Weisser et al., 2019), TSSOS (Wang et al., 2021b), Chordal decomposition (Wang et al., 2021a)) on four challenging configurations. Experiments were run on an Intel Xeon Gold 6246 (16 cores, 512GB RAM). Our approach achieves substantial speedups: $21\times$ faster than SoS.jl and $15\times$ faster than TSSOS on 6 variables with degree 20, and over $2000\times$ faster on 8 variables with degree 20. For the largest instances (6 variables with degree 40 and 100 variables with degree 10), all baseline solvers encounter out-of-memory errors or timeouts, while our method succeeds in under 60 seconds.

For distribution shift evaluation, the Transformer model maintains relatively strong performance even on degrees unseen during training. This robustness likely stems from the monomial-embedding technique, which captures structural patterns that generalize beyond the training distribution. Despite being trained only on degree 12 polynomials, our model can successfully predict bases for degree 20 polynomials by leveraging familiar substructures—though it has not seen complete degree 20 monomials, it has encountered their constituent parts (e.g., isolated variables like $x_1$). This compositional generalization has been similarly observed (Kera et al., 2025b).

Table 2: Evaluation on large configurations. Entries are average time (s). "–" = OOM/timeout.

|  | **Ours** | **SoS.jl** | **TSSOS** | **Chordal** |
|---|---|---|---|---|
| 6 vars, deg 20 | 5.64 | 119.98 | 86.53 | 105.54 |
| 6 vars, deg 40 | 42.8 | – | – | – |
| 8 vars, deg 20 | 1.46 | 3037.85 | 2674.50 | 3452.98 |
| 100 vars, deg 10 | 18.3 | – | – | – |

## 5 THEORETICAL ANALYSIS

In this section, we establish elementary guarantees for our approach. Throughout our analysis, we assume that the computational cost of solving an SDP with basis size $m$ scales as $\Theta(m^\omega)$, where the exponent $\omega$ depends on the specific solver implementation. While the practical computational complexity of solving SDPs also depends on the number of matrix constraints, we empirically observe that for the sparse polynomial problems we consider, the constraint count scales linearly with the basis size $m$.

The standard SOS certification approach constructs a single large SDP using all monomials in the half Newton polytope $\frac{1}{2}N(p)$, incurring computational cost $\Theta(m^\omega)$ where $m = \left|\frac{1}{2}N(p)\right|$. Our method, by contrast, solves a sequence of progressively larger SDPs with basis sizes $m_1, m_2, \ldots, m_j$, terminating upon finding a feasible SOS decomposition. The total computational cost under our approach is $\Theta\left(\sum_{i=1}^{j} m_i^\omega\right)$, which can be substantially smaller than the standard cost when accurate predictions enable early termination with compact bases.

Given a polynomial $p$, we define a *prediction* as specifying: (i) an ordering $\nu = (u_1, \ldots, u_N)$ over the half Newton polytope $\frac{1}{2}N(p)$, and (ii) a schedule $\sigma = (m_1 < \cdots < m_k)$ of basis sizes to attempt. We measure the prediction quality through the *coverage rank* $\eta$, defined as the smallest index $j$ such that $\{u_1, \ldots, u_j\}$ is a monomial SOS basis for the polynomial $p$.

**Remark 3.** *This cost model provides several straightforward insights. In the best case, when the coverage rank equals the minimal basis size and $B_{cov}$ from Algorithm 1 is minimal, our method finds the certificate in a single SDP solve at minimal cost, which we formalize in Lemma 5. In the worst case, when the coverage rank equals $\left|\frac{1}{2}N(p)\right|$, our method requires $k$ SDP solves with basis sizes $m_1, \ldots, m_k$, incurring at most $O(k)$ times the cost of the standard approach, which we formally state in Lemma 6.*

We conclude with a natural choice of expansion schedule, which is to increase the basis size by a constant factor $\rho > 1$ at each iteration and relate this to the coverage rank.

**Lemma 4** (Geometric Expansion). *Let $B_{cov}$ be a coverage-repaired SOS basis obtained by Algorithm 1. Let the coverage rank be $\eta$, set $m_1 = |B_{cov}|$ and choose $\rho > 1$. Then, Algorithm 2 performs at most $1 + \lceil \log_\rho(\eta/m_1) \rceil$ SDP solves and has total cost at most $\Theta\left(\frac{\rho^\omega}{1 - \rho^{-\omega}} \eta^\omega\right)$.*

We refer to Theorem 9 for an ERM approach to select the parameter $\rho$ for the geometric expansion schedule that minimizes the expected cost of the SDP solves.

## 6 CONCLUSION AND LIMITATIONS

This paper introduces a learning-augmented algorithm for SOS programming that achieves significant speedups while maintaining theoretical guarantees. By framing basis selection as a sequence prediction task, we bridge the gap between practical efficiency and mathematical rigor. Our approach demonstrates robustness across different polynomial families and variable counts, suggesting the learned patterns generalize beyond the training distribution. The systematic repair mechanism enables the algorithm to recover from initial prediction errors, while the Transformer model learns to exploit sparsity patterns that traditional methods miss. These capabilities combine to deliver consistent performance improvements across diverse problem instances.

Nonetheless, our approach has limitations. Our extensive experiments, comprising 100 million polynomials across 200 synthetic datasets, provide broad coverage of polynomial structures. However, the lack of widely available real-world SOS benchmarks requires us to rely on synthetic polynomial generation for empirical evaluation. More broadly, our predictor may be sensitive to non-canonical choices in representing $p$ (beyond variable permutations), and addressing this via deterministic test-time canonicalization is an interesting direction for future work (cf. RECON (Urbano et al., 2026)). Furthermore, although our method demonstrates strong performance in the studied regime, scalability remains a challenge: going beyond 100 variables or tackling problems with many more terms is difficult due to the Transformer's limited context length and the computational bottleneck posed by solving large SDPs. Real-world SOS instances with sparse structure and fixed coefficients are also rare, further constraining empirical assessment; while we include one control example in the appendix (Appendix E.11), large-scale realistic benchmarks remain an open need.

## ACKNOWLEDGEMENTS

This research was partially supported by the Deutsche Forschungsgemeinschaft (DFG) through the DFG Cluster of Excellence MATH+ (EXC-2046/1, EXC-2046/2, project id 390685689), as well as by the German Federal Ministry of Research, Technology and Space (research campus Modal, fund number 05M14ZAM, 05M20ZBM) and the VDI/VDE Innovation + Technik GmbH (fund number 16IS23025B). D. Martínez-Rubio was partially funded by grant PID2024-160448NA-I00 and by La Caixa Junior Leader Fellowship 2025.

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

# A  STATEMENTS

## A.1  USE OF LARGE LANGUAGE MODELS

Large language models were used to aid in writing (polishing text), retrieval of related work, generating code for plots, and implementing standard components. No novel research ideas or results were produced by LLMs.

# B  IMPLEMENTATION DETAILS

## B.1  SOFTWARE

We implemented our approach using a combination of Python and Julia. Table 3 lists the key software packages and their versions used in our implementation.

Table 3: Software packages and versions used in our implementation.

| Package | Version |
|---|---|
| Python | 3.9.19 |
| Julia | 1.11.5 |
| Mosek | 11.0.22 |
| SumOfSquares (Julia) | 0.7.4 |
| CVXPY | 1.6.5 |
| SCS | 3.2.7 |
| NumPy | 2.0.2 |

Experiments were run on an Intel Xeon Gold 6246 (16 cores, 512GB RAM) and an NVIDIA A40 (48GB RAM).

## B.2  MODEL TRAINING

We use a standard encoder-decoder Transformer with 6 layers each, model dimension 512, and 8 attention heads. Key training hyperparameters: batch size 128, learning rate $1 \times 10^{-4}$, AdamW optimizer with weight decay $1 \times 10^{-4}$, and 1 epoch. The model jointly trains basis selection (cross-entropy loss) and coefficient prediction (MSE loss).

Table 4: Transformer model configuration for Grid 1, 2, and 3.

| Parameter | Value |
|---|---|
| Model dimension ($d_{\mathrm{model}}$) | 512 |
| Attention heads | 8 |
| Encoder/decoder layers | 6 each |
| Batch size | 128 |
| Learning rate | $1 \times 10^{-4}$ |
| Optimizer | AdamW |
| Weight decay | $1 \times 10^{-4}$ |
| Training epochs | 1 |

### B.2.1  TRAINING TIME

For the Transformer models in Table 1, the average training time is approximately 5 hours, with a maximum of under 6 hours for the dense dataset. For the harder datasets in Table 2, the average training time is approximately 10 hours, with the maximum remaining under 11 hours. Note that for smaller instances, the training time is substantially higher than the time required to solve the instances directly. In contrast, for the larger instances, traditional SOS solvers either time out after 2 hours or require more than 60 minutes, indicating that the training cost is effectively amortized over

Table 5: Transformer model configuration for Grid 4.

| Parameter | Value |
|---|---|
| Model dimension ($d_{\text{model}}$) | 1024 |
| Attention heads | 16 |
| Encoder layers | 12 |
| Decoder layers | 8 |
| Batch size | 8 |
| Learning rate | $1 \times 10^{-4}$ |
| Optimizer | AdamW |
| Weight decay | $1 \times 10^{-4}$ |
| Warmup ratio | 0.05 |
| Training epochs | 1 |

Table 6: Transformer model configuration for experiments 100 variables.

| Parameter | Value |
|---|---|
| Model dimension ($d_{\text{model}}$) | 768 |
| Attention heads | 12 |
| Encoder/decoder layers | 6 each |
| Batch size | 64 |
| Learning rate | $1 \times 10^{-4}$ |
| Optimizer | AdamW |
| Weight decay | $1 \times 10^{-4}$ |
| Training epochs | 1 |

a small number of instances. In Appendix E.12, we provide an amortization analysis of the training cost and show when the one-time learning expense is outweighed by per-instance solver speedups.

## C  ADDITIONAL FIGURES

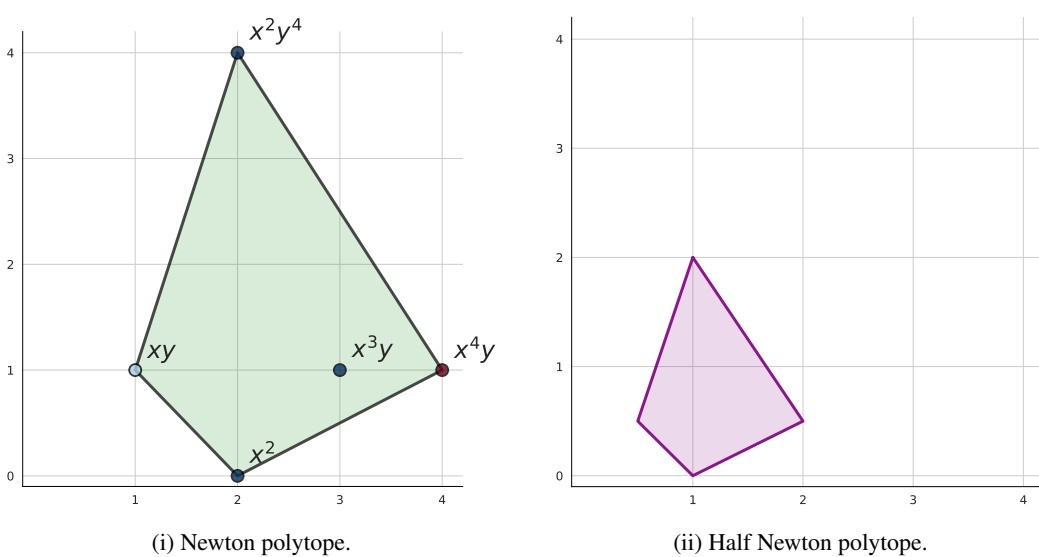

(i) Newton polytope.  (ii) Half Newton polytope.

Figure 6: Comparison of Newton polytope and half Newton polytope for the polynomial $p(\mathbf{x}) = 7x^4y + x^3y + x^2y^4 + x^2 + 3xy$. Support of the polynomial is annotated in Figure 6i. On the right, we show the half Newton polytope, which is the scaled down version of the Newton polytope. If an SOS decomposition exists, then the half Newton polytope must contain a basis for the polynomial.

# D  DATASET GENERATION

We train our models on three data grids totaling 205 datasets, each containing 1M polynomials split into 99% training, 0.5% test, and 0.5% validation sets on the following grids, generated with a fixed random seed.

**Grid 1:** 8 variables, degrees $\{4, 6, 8, 10, 12, 14, 16, 18, 20\}$, average basis sizes $\{10, 20, 30\}$, matrix structures $\{$sparse, low-rank, block diagonal, full-rank dense$\}$ (108 datasets).

**Grid 2:** Variables $\{4, 6, 8, 10, 12, 14, 16, 18, 20\}$, degree 12, average basis size 30, matrix structures $\{$sparse, low-rank, block diagonal, full-rank dense$\}$ (36 datasets).

**Grid 3:** 4 variables, degree 6, average basis size 20, matrix structures $\{$sparse, low-rank, block diagonal, full-rank dense$\}$ (4 datasets).

**Grid 4:** 6 variables, degree 20, average basis size 60, matrix structures $\{$sparse, low-rank, block diagonal, full-rank dense$\}$ (4 datasets).

**Large Scale Special:** In addition, we generated a dataset with 100 variables, degree 10, average basis size 20, matrix structures blockdiagonal. We also constructed a dataset with 6 variables, degree 40, average basis size 40, matrix structures full-rank dense, with the property that all basis monomials have only even degree in all variables. This is to make it more challenging for the model to predict the basis. (2 datasets).

For out-of-distribution experiments, we partly reused the datasets above, i.e., using the model trained on only degree 12 monomials on the degrees $\{4, 6, 8, 10, 12, 14, 16, 18, 20\}$. In addition, we generated datasets with 1000 test instances in the following grids.

**Non-SOS Grid:** $n = 4$, degree 6, 20 monomials; $n = 6$, degree 12, 30 monomials; $n = 6$, degree 20, 60 monomials. We construct non-SOS polynomials by performing eigenvalue decomposition of PSD Gram matrices and then perturbing $10-20\%$ of the eigenvalues to be negative while preserving the overall matrix structure. (3 datasets).

**Permutation Grid:** 8 variables, block diagonal matrices with degree 8 and average basis sizes $\{40, 50\}$; dense full-rank matrices with degree 9 and average basis size 40; dense full-rank matrices with degree 7 and average basis size 50. These were used for ablation studies on the number of permutations. (4 datasets).

**OOD Grid:** 6 variables, degrees $\{10, 12, 14\}$, average basis sizes $\{20, 25, 35, 40\}$, matrix structures $\{$sparse, low-rank, block diagonal, full-rank dense$\}$ (48 new datasets).

This gives a total of $108 + 36 + 4 + 4 + 3 + 48 + 2 = 205$ datasets.

In principle, there is no need to generate these datasets, as one can generate them on the fly. For our experiments, we generated them to avoid any potential issues with the generation process. Our largest dataset (dense full-rank matrices with degree 10 and average basis size 60 and 6 variables) required 2 hours to generate, while most other datasets from Grid 1 took less then 10 minutes to generate on a 8 core CPU.

## D.1  POLYNOMIAL SAMPLING

For each polynomial in our datasets, we sample monomials uniformly over the space of all monomials up to the given degree $d$. Our sampling procedure operates in two stages: (i) **Degree sampling:** We first sample the degree $k$ of each monomial, where $k \leq d$. To ensure uniform distribution over all monomials, we weight the degree selection by the number of monomials of each degree, which is $\binom{n+k-1}{k}$ for $n$ variables and degree $k$. (ii) **Monomial sampling:** Given the selected degree $k$, we uniformly sample a monomial of degree $k$ from the set of all such monomials.

# E  ADDITIONAL EXPERIMENTS

## E.1  COST OF FAILED SDP SOLVES AND NON-SOS POLYNOMIALS

A key component of our approach is the ability to recover from failed SDP solves. However, in theory, the cost of a failed SDP solve can be quite high, even for a geometric expansion schedule, due to a potentially large number of SDP solves. We show that in practice, the cost of a failed SDP solve is not as high as expected.

We choose $\rho = 1.5$ and $L = 4$ on the out-of-distribution datasets from the ablation study on permutations. Here we only consider instances, where our method has to fallback to the Newton polytope. For these instances, we compute the time-weighted runtime ratio, which is the ratio of the runtime of the combined runtime of the SDPs and the final SDP based on the Newton polytope. Across all configurations, the time-weighted runtime ratio is very close to 1, indicating that the cost of a failed SDP solve is not as high as expected.

Table 7: Time-weighted runtime ratio by configuration.

| Configuration | Config 1 | Config 2 | Config 3 | Config 4 | Config 5 | Config 6 |
|---|---|---|---|---|---|---|
| Runtime Ratio | 1.069 | 1.068 | 1.097 | 1.009 | 1.003 | 1.004 |

We conjecture that this favorable runtime behavior is actually caused by SDP problems with small bases being not only infeasible but strongly infeasible, which can potentially be detected much faster by interior point solvers (Pólik & Terlaky, 2009).

While our approach accelerates the computation of positive SOS certificates, it cannot accelerate the computation of negative SOS certificates. This is because proving non-existence of an SOS decomposition requires considering the full Newton polytope. Our approach therefore incurs additional overhead for non-SOS polynomials compared to the standard Newton polytope approach. To assess the impact of this overhead, we construct non-SOS polynomials by performing eigenvalue decomposition of PSD Gram matrices and then perturbing the eigenvalues, making approximately $10 - 20\%$ of them negative while preserving the overall matrix structure, creating polynomials that are subtly non-SOS.

We compute the computational overhead as the ratio of the runtime of the combined runtime of the SDPs and the final SDP based on the Newton polytope, using $\rho = 1.5$ and $L = 4$.

Table 8: Overhead for non-SOS polynomials relative to standard Newton polytope approach.

| Dataset | Runtime Ratio |
|---|---|
| $n = 4$, degree 6, 20 monomials | 2.24 |
| $n = 6$, degree 12, 30 monomials | 1.76 |
| $n = 6$, degree 20, 60 monomials | 1.05 |

The overhead decreases for larger problem instances, ranging from $2.24\times$ for small problems to only $1.05\times$ for larger ones, suggesting that the computational penalty for non-SOS cases becomes negligible as problem size increases.

## E.2  ABLATION: NUMBER OF PERMUTATIONS

We investigate the impact of the number of permutations $L \in \{1, 2, 4, 6, 8, 10, 12\}$ on our repair mechanism's performance. We evaluate our method on out-of-distribution test cases where the model encounters polynomials with different characteristics than those seen during training. This distribution shift causes the initial predictions to be less accurate, making the repair mechanism necessary and allowing us to study its effectiveness.

Specifically, we use a Transformer trained on polynomials with 6 variables, degree 12, average basis size 30, and sparse matrix structure, and test it on six different configurations, which were selected based on the criterion that initial success rates (after greedy repair) were below 50%:

- **Configuration 1:** Block diagonal matrices, degree 8, basis size 40

- **Configuration 2:** Block diagonal matrices, degree 8, basis size 50

- **Configuration 3:** Block diagonal matrices, degree 9, basis size 30

- **Configuration 4:** Dense full-rank matrices, degree 9, basis size 40

- **Configuration 5:** Dense full-rank matrices, degree 7, basis size 50

- **Configuration 6:** Dense full-rank matrices, degree 10, basis size 30

All configurations maintain 6 variables while varying degree, matrix structure, and basis size. These configurations test the model's ability to generalize across different polynomial complexities and underlying matrix structures.

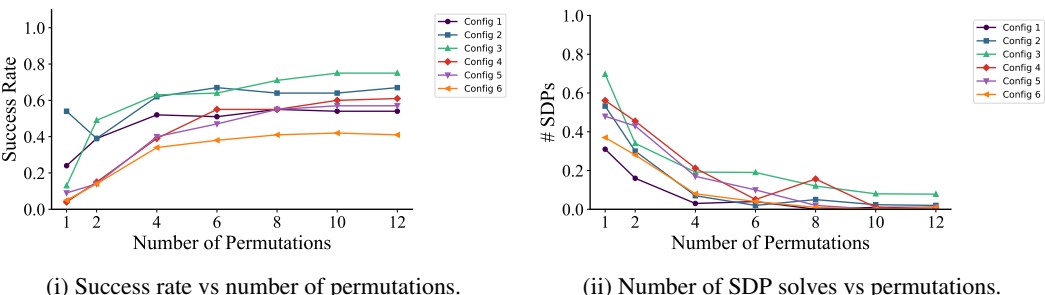

(i) Success rate vs number of permutations.  (ii) Number of SDP solves vs permutations.

Figure 7: Performance metrics as a function of permutation count.

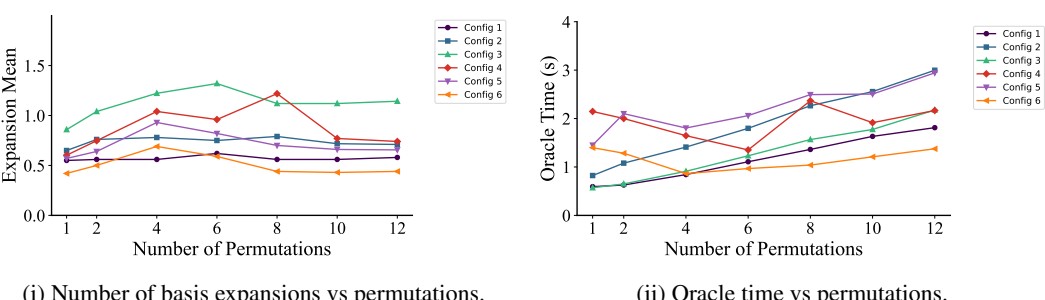

(i) Number of basis expansions vs permutations.  (ii) Oracle time vs permutations.

Figure 8: Computational overhead analysis for varying permutation counts.

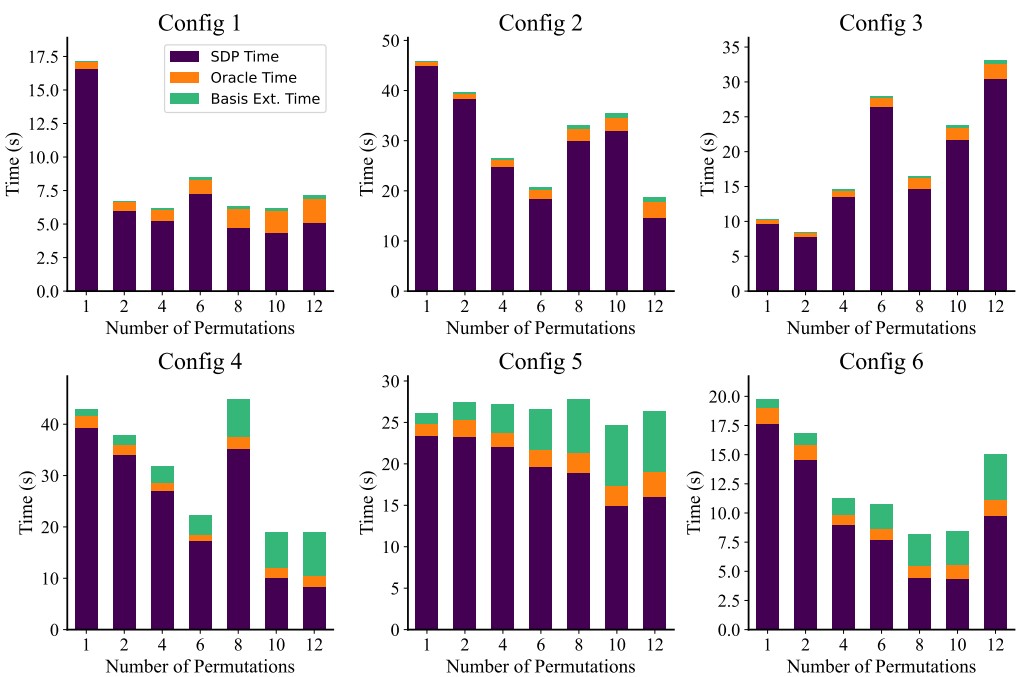

(i) Runtime allocation for configurations and permutations.

Figure 9: Basis expansion behavior with varying permutation counts.

**Key Results:** The ablation study reveals several important trends: (1) Success rates improve substantially with increasing permutations, with diminishing returns beyond $L = 8$; (2) Transformer inference time scales approximately linearly with the number of permutations, as expected; (3) The number of SDP solves decreases with more permutations, likely due to better scoring; (4) Fallback to Newton polytope methods decreases as more permutations increase the probability of finding a valid basis; (5) The average number of basis expansions remains relatively constant, indicating that additional permutations primarily improve success rate rather than requiring more expansions per successful case.

Figure 9 shows that the runtime is typically dominated by SDP solves. The optimal number of permutations is usually around $L = 4 - 8$, where the success rate plateaus and further increases in permutations mainly increase the inference time.

Empirically, this plateauing effect motivates our choice of $L = 4$–$8$ as a practical default. For clarity, consider an example from the sparse setting in Table 1: the true minimal basis contains 61 monomials. Taking the union of $L = 4$ permutations produces a candidate set with 78 monomials; $L = 8$ gives 82; by $L = 16$, still only 82. In contrast, the full Newton polytope contains 242 monomials. Thus, additional permutations rapidly saturate the set—extra candidates rarely appear beyond $L = 8$. This demonstrates that moderate values of $L$ reliably yield compact, near-minimal candidate bases, while avoiding the computational cost associated with much larger sets.

### E.3    ABLATION: EXPANSION SCHEDULE

We ablate the expansion factor $\rho \in \{1.1, 1.2, 1.4, 1.6, 1.8, 2.0\}$ while keeping all other hyperparameters fixed. Compared to the permutation count, $\rho$ has a smaller effect on overall performance.

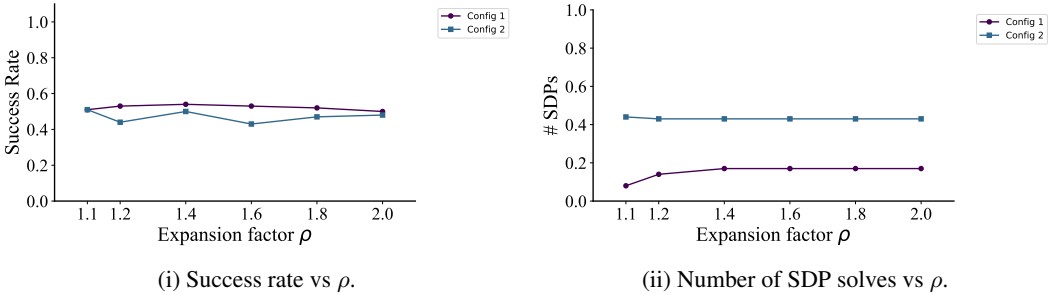

(i) Success rate vs $\rho$.
(ii) Number of SDP solves vs $\rho$.

Figure 10: Performance metrics as a function of the expansion factor.

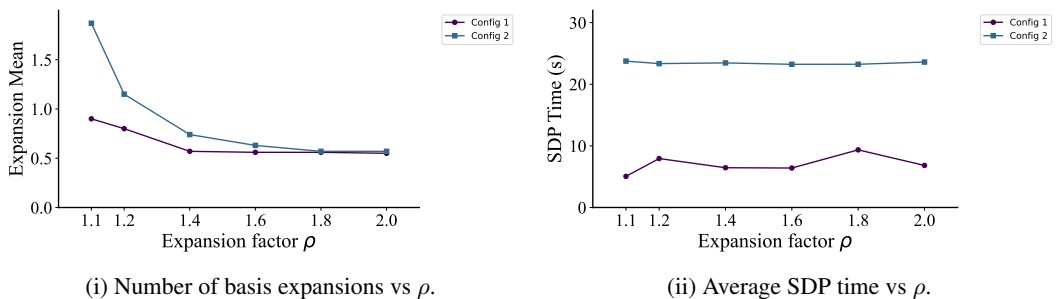

(i) Number of basis expansions vs $\rho$.
(ii) Average SDP time vs $\rho$.

Figure 11: Expansion behavior and solver time across expansion factors.

**Key Results:** The expansion factor is less critical than the number of permutations. Larger $\rho$ reduces the number of basis expansions, as expected, while SDP time remains fairly consistent across settings, likely due to the large gap between the Newton polytope size and the initially proposed basis. Overall, slightly smaller expansion factors (e.g., $\rho \leq 1.4$) are marginally more favorable, though differences are modest.

### E.4 BASIS EXTENSION ALGORITHMS

The basis extension mechanism described in Section 3.3 employs two key algorithms. Algorithm 3 presents the main extension algorithm, which iteratively adds monomials to the basis until it can theoretically represent the SOS decomposition of the polynomial. At each iteration, it computes which monomials from the support $S$ are missing from the current basis products, then selects the most promising candidate to add.

The algorithm calls the subroutine FIND-CANDIDATE to select the most promising monomial to add to the basis (Algorithm 4). Given a set of missing monomials $M$ and current basis $B$, the subroutine counts how many times each candidate monomial $d$ appears as a divisor of monomials in $M$, then returns the most frequent candidate.

### E.5 BASIS EXTENSION ANALYSIS

We analyze the performance of the basis extension mechanism in the context of the additive combinatorics inverse problem. We test our ability to recover the complete set $A$ from a partial observation $A'$ and the complete sumset $A + A$.

For this experiment, we generate a polynomial with a known basis $A$ and then remove $k$ elements from the basis to create a partial observation $A'$. We then use our extension mechanism to attempt to recover the missing elements and reconstruct $A$. Finally, we measure whether we were able to recover the original set $A$ exactly (equal), the original set $A$ is a subset of the recovered set (superset), or the recovered set does not contain the original set $A$ (insufficient).

---

**Algorithm 3** COVERAGEREPAIR

---

**Input:** Initial basis $B_0$, polynomial $p$, maximum iterations $max\_iter$
**Output:** Extended basis $B$

1: $B \leftarrow B_0$
2: $S \leftarrow \text{support}(p)$
3: **for** $i = 1$ to $max\_iter$ **do**
4:     $P \leftarrow \{m_1 \cdot m_2 \mid m_1, m_2 \in B\}$
5:     $M \leftarrow S \setminus P$
6:     **if** $M = \emptyset$ **then**
7:       **return** $B$
8:     **else**
9:       $d^* \leftarrow \text{FIND-CANDIDATE}(M, B)$
10:    **end if**
11:    **if** $d^* = \text{None}$ **then**
12:       **return** $B$
13:    **else**
14:       $B \leftarrow B \cup \{d^*\}$
15:    **end if**
16: **end for**
17: **return** $B$

---

**Algorithm 4** FIND-CANDIDATE$(M, B)$

---

**Input:** Set of missing monomials $M$, current basis $B$
**Output:** Monomial $d^*$ to add to the basis

1: $D \leftarrow$ empty counter
2: **for** each $m \in M$ **do**
3:     **for** each $b \in B$ **do**
4:       **if** $m$ is divisible by $b$ **then**
5:         $d \leftarrow m/b$
6:         Increment $D[d]$ by 1
7:       **end if**
8:     **end for**
9: **end for**
10: **if** $D$ is empty **then**
11:    **return** None
12: **end if**
13: $d^* \leftarrow \arg\max_{d \in D} D[d]$
14: **return** $d^*$

---

| $k$ | Equal | Superset | Insufficient |
|---|---|---|---|
| 1 | 999 | 0 | 1 |
| 2 | 992 | 2 | 6 |
| 3 | 982 | 2 | 16 |
| 4 | 942 | 24 | 34 |
| 5 | 801 | 67 | 132 |
| 6 | 695 | 13 | 292 |

Table 9: Basis extension analysis in the additive combinatorics setting. The table shows how many times the original set $A$ was recovered exactly (equal), the original set $A$ is a subset of the recovered set (superset), or the recovered set does not contain the original set $A$ (insufficient) for different values of $k$ missing elements from the partial observation $A'$.

The results show that the basis extension mechanism is able to recover the original set $A$ for most cases if $k \leq 4$. This demonstrates the effectiveness of our greedy approach to the additive combinatorics inverse problem: given $A' \subseteq A$ and $A + A$, we can reliably recover $A$ when the partial observation $A'$ contains most of the elements of $A$. Removing more elements results in a higher number of insufficient cases, highlighting the fundamental difficulty of the inverse problem when the partial observation becomes too sparse.

### E.5.1 FALSE POSITIVES ANALYSIS

We also analyze the impact of false positives on the extension mechanism. In the additive combinatorics setting, this corresponds to the scenario where the sumset $A + A$ contains some erroneous elements that are not actually part of the true sumset.

We generate a polynomial with a known basis $A$ and then remove $k$ elements from the basis to create $A'$. Then we add the same number of random elements to the sumset $A + A$ (simulating false positives) and measure whether we can recover exactly the original set $A$ plus the elements that would generate the false positive sumset elements (exact), the recovered set contains additional unnecessary elements (superset), or the recovered set is insufficient to cover the original set $A$ (insufficient).

| $k$ | Exact | Superset | Insufficient |
|---|---|---|---|
| 1 | 991 | 2 | 7 |
| 2 | 980 | 20 | 0 |
| 3 | 957 | 43 | 0 |
| 4 | 902 | 98 | 0 |
| 5 | 745 | 205 | 50 |
| 6 | 640 | 154 | 206 |

Table 10: Analysis of basis extension with false positives in the additive combinatorics setting. The table shows how many times we recovered exactly the original set $A$ plus the elements needed to generate the false positive sumset elements (exact), recovered a superset containing additional unnecessary elements (superset), or failed to recover the complete set $A$ (insufficient) for different values of $k$ missing elements from the partial observation $A'$.

The results show that the extension mechanism is able to recover the original set $A$ for the majority of cases, adding no additional elements beyond those necessary to explain the false positive elements in the sumset. This demonstrates the robustness of our greedy approach to the additive combinatorics inverse problem even in the presence of noise in the sumset data.

### E.6 EMPIRICAL BASIS SIZE ANALYSIS

To demonstrate that the theoretical lower bounds are often tight in practice, we conducted extensive empirical analysis across multiple polynomial structures and problem sizes. This analysis validates that our training data generation produces near-minimal bases, supporting the key message that these bounds are often tight and therefore we trained the model on near minimal basis.

The analysis builds on two key theoretical results established in Lemma 7 and Lemma 8. The combinatorial lower bound ($\text{LB}_c$) establishes that for any SOS basis $B$ of polynomial $p$ with support $S(p)$, we have $|B| \geq \frac{\sqrt{1+8|S(p)|}-1}{2}$. This bound arises from the fact that a basis of $m$ monomials can generate at most $\frac{m(m+1)}{2}$ distinct monomials through pairwise products. The Newton polytope vertices lower bound ($\text{LB}_v$) provides a geometric constraint, requiring that certain vertices of the half Newton polytope must appear in any valid basis.

We conducted systematic experiments across four polynomial matrix structures: block-diagonal matrices with sparse block patterns, low-rank matrices with reduced rank structure, dense full-rank matrices with general structure, and sparse matrices with many zero entries. For each structure, we tested multiple configurations with problem sizes $n \in \{4, 6\}$ variables, degrees $d \in \{3, 6, 10\}$, and target basis sizes $B^* \in \{20, 30, 60\}$.

Each table reports metrics for measuring lower bound tightness. $B^*$ denotes the actual basis size used to generate the polynomial. $\text{LB}_v$ and $\text{LB}_c$ represent the Newton polytope vertices and combinatorial lower bounds respectively. $|\frac{1}{2}N(p)|$ gives the size of the half Newton polytope used in traditional approaches. $\Delta^{(v)} = B^* - \text{LB}_v$ and $\rho^{(v)} = B^*/\text{LB}_v$ measure the gap and ratio between actual basis size and vertex bound, while $\Delta^{(c)}$ and $\rho^{(c)}$ provide analogous metrics for the combinatorial bound.

The results demonstrate that the Newton polytope vertices bound is remarkably tight across all polynomial structures. For low-rank and dense matrices, we observe $\Delta^{(v)} = 0$ (perfect tightness) in most cases, with $\rho^{(v)} = 1.000$. For block-diagonal and sparse matrices, gaps remain small with $\Delta^{(v)} \leq 4$ and $\rho^{(v)} \leq 1.065$. The combinatorial bound shows varying tightness depending on structure, achieving $\rho^{(c)} \approx 1.000 - 1.030$ for structured matrices but $\rho^{(c)} \approx 1.5 - 2.5$ for sparse structures. Importantly, the half Newton polytope $|\frac{1}{2}N(p)|$ is significantly larger than actual basis sizes, containing 3-15× more monomials than necessary and demonstrating substantial redundancy in traditional approaches.

Table 11: Lower bounds for blockdiagonal matrices with $n = 6$ variables and degree $d = 10$.

| $B^*$ | $\text{LB}_v$ | $\text{LB}_c$ | $|\frac{1}{2}N(p)|$ | $\Delta^{(v)}$ | $\rho^{(v)}$ | $\Delta^{(c)}$ | $\rho^{(c)}$ |
|---|---|---|---|---|---|---|---|
| 60 | 60 | 32 | 299 | 0 | 1.000 | 28 | 1.875 |
| 66 | 62 | 33 | 339 | 4 | 1.065 | 33 | 2.000 |
| 56 | 55 | 31 | 189 | 1 | 1.018 | 25 | 1.806 |
| 55 | 54 | 31 | 240 | 1 | 1.019 | 24 | 1.774 |
| 60 | 60 | 32 | 248 | 0 | 1.000 | 28 | 1.875 |
| 66 | 64 | 33 | 281 | 2 | 1.031 | 33 | 2.000 |
| 55 | 55 | 31 | 211 | 0 | 1.000 | 24 | 1.774 |
| 56 | 53 | 31 | 295 | 3 | 1.057 | 25 | 1.806 |

Table 12: Lower bounds for low-rank matrices with $n = 6$ variables and degree $d = 10$.

| $B^*$ | $LB_v$ | $LB_c$ | $|\frac{1}{2}N(p)|$ | $\Delta^{(v)}$ | $\rho^{(v)}$ | $\Delta^{(c)}$ | $\rho^{(c)}$ |
|---|---|---|---|---|---|---|---|
| 73 | 73 | 71 | 252 | 0 | 1.000 | 2 | 1.028 |
| 70 | 70 | 69 | 325 | 0 | 1.000 | 1 | 1.014 |
| 63 | 63 | 62 | 250 | 0 | 1.000 | 1 | 1.016 |
| 64 | 64 | 63 | 215 | 0 | 1.000 | 1 | 1.016 |
| 74 | 74 | 73 | 378 | 0 | 1.000 | 1 | 1.014 |
| 65 | 65 | 64 | 292 | 0 | 1.000 | 1 | 1.016 |
| 69 | 69 | 68 | 331 | 0 | 1.000 | 1 | 1.015 |
| 72 | 72 | 71 | 316 | 0 | 1.000 | 1 | 1.014 |
| 59 | 59 | 59 | 254 | 0 | 1.000 | 0 | 1.000 |

Table 13: Lower bounds for dense full-rank matrices with $n = 6$ variables and degree $d = 10$.

| $B^*$ | $LB_v$ | $LB_c$ | $|\frac{1}{2}N(p)|$ | $\Delta^{(v)}$ | $\rho^{(v)}$ | $\Delta^{(c)}$ | $\rho^{(c)}$ |
|---|---|---|---|---|---|---|---|
| 48 | 48 | 48 | 243 | 0 | 1.000 | 0 | 1.000 |
| 58 | 58 | 57 | 227 | 0 | 1.000 | 1 | 1.018 |
| 74 | 74 | 73 | 378 | 0 | 1.000 | 1 | 1.014 |
| 73 | 73 | 71 | 252 | 0 | 1.000 | 2 | 1.028 |
| 70 | 70 | 69 | 325 | 0 | 1.000 | 1 | 1.014 |
| 68 | 68 | 67 | 331 | 0 | 1.000 | 1 | 1.015 |
| 59 | 59 | 59 | 254 | 0 | 1.000 | 0 | 1.000 |
| 63 | 63 | 63 | 250 | 0 | 1.000 | 0 | 1.000 |
| 47 | 47 | 47 | 175 | 0 | 1.000 | 0 | 1.000 |

Table 14: Lower bounds for sparse matrices with $n = 6$ variables and degree $d = 10$.

| $B^*$ | $LB_v$ | $LB_c$ | $|\frac{1}{2}N(p)|$ | $\Delta^{(v)}$ | $\rho^{(v)}$ | $\Delta^{(c)}$ | $\rho^{(c)}$ |
|---|---|---|---|---|---|---|---|
| 56 | 56 | 32 | 189 | 0 | 1.000 | 24 | 1.750 |
| 58 | 56 | 30 | 235 | 2 | 1.036 | 28 | 1.933 |
| 61 | 61 | 42 | 288 | 0 | 1.000 | 19 | 1.452 |
| 55 | 54 | 33 | 211 | 1 | 1.019 | 22 | 1.667 |
| 66 | 64 | 36 | 281 | 2 | 1.031 | 30 | 1.833 |
| 66 | 65 | 36 | 339 | 1 | 1.015 | 30 | 1.833 |
| 56 | 55 | 32 | 295 | 1 | 1.018 | 24 | 1.750 |
| 67 | 66 | 35 | 280 | 1 | 1.015 | 32 | 1.914 |
| 62 | 62 | 42 | 264 | 0 | 1.000 | 20 | 1.476 |
| 57 | 57 | 34 | 212 | 0 | 1.000 | 23 | 1.676 |

Table 15: Lower bound tightness for block diagonal matrices, $n = 4$, $d = 3$, average sampled basis size $B^* = 20$.

| $B^*$ | $LB_v$ | $LB_c$ | $|\frac{1}{2}N(p)|$ | $\Delta^{(v)}$ | $\rho^{(v)}$ | $\Delta^{(c)}$ | $\rho^{(c)}$ |
|---|---|---|---|---|---|---|---|
| 28 | 14 | 11 | 29 | 14 | 2.000 | 17 | 2.545 |
| 19 | 18 | 10 | 24 | 1 | 1.056 | 9 | 1.900 |
| 22 | 16 | 10 | 24 | 6 | 1.375 | 12 | 2.200 |
| 11 | 11 | 7 | 15 | 0 | 1.000 | 4 | 1.571 |
| 13 | 12 | 8 | 17 | 1 | 1.083 | 5 | 1.625 |
| 25 | 16 | 11 | 28 | 9 | 1.562 | 14 | 2.273 |
| 14 | 13 | 8 | 16 | 1 | 1.077 | 6 | 1.750 |
| 15 | 14 | 8 | 24 | 1 | 1.071 | 7 | 1.875 |
| 21 | 15 | 10 | 25 | 6 | 1.400 | 11 | 2.100 |
| 25 | 13 | 11 | 28 | 12 | 1.923 | 14 | 2.273 |

Table 16: Lower bound tightness for low rank matrices, $n = 4$, $d = 3$, average sampled basis size $B^* = 20$.

| $B^*$ | $\mathbf{LB}_v$ | $\mathbf{LB}_c$ | $\left|\frac{1}{2}N(p)\right|$ | $\Delta^{(v)}$ | $\rho^{(v)}$ | $\Delta^{(c)}$ | $\rho^{(c)}$ |
|---|---|---|---|---|---|---|---|
| 11 | 11 | 11 | 15 | 0 | 1.000 | 0 | 1.000 |
| 19 | 18 | 16 | 24 | 1 | 1.056 | 3 | 1.188 |
| 22 | 22 | 17 | 24 | 0 | 1.000 | 5 | 1.294 |
| 13 | 13 | 13 | 17 | 0 | 1.000 | 0 | 1.000 |
| 28 | 25 | 19 | 29 | 3 | 1.120 | 9 | 1.474 |
| 25 | 24 | 18 | 28 | 1 | 1.042 | 7 | 1.389 |
| 15 | 15 | 14 | 24 | 0 | 1.000 | 1 | 1.071 |
| 14 | 14 | 13 | 16 | 0 | 1.000 | 1 | 1.077 |
| 21 | 20 | 17 | 25 | 1 | 1.050 | 4 | 1.235 |
| 13 | 13 | 12 | 15 | 0 | 1.000 | 1 | 1.083 |

Table 17: Lower bound tightness for dense matrices, $n = 4$, $d = 3$, average sampled basis size $B^* = 20$.

| $B^*$ | $\mathbf{LB}_v$ | $\mathbf{LB}_c$ | $\left|\frac{1}{2}N(p)\right|$ | $\Delta^{(v)}$ | $\rho^{(v)}$ | $\Delta^{(c)}$ | $\rho^{(c)}$ |
|---|---|---|---|---|---|---|---|
| 11 | 11 | 11 | 15 | 0 | 1.000 | 0 | 1.000 |
| 19 | 18 | 16 | 24 | 1 | 1.056 | 3 | 1.188 |
| 13 | 13 | 13 | 17 | 0 | 1.000 | 0 | 1.000 |
| 22 | 22 | 17 | 24 | 0 | 1.000 | 5 | 1.294 |
| 28 | 25 | 19 | 29 | 3 | 1.120 | 9 | 1.474 |
| 25 | 24 | 18 | 28 | 1 | 1.042 | 7 | 1.389 |
| 21 | 20 | 17 | 25 | 1 | 1.050 | 4 | 1.235 |
| 14 | 14 | 13 | 16 | 0 | 1.000 | 1 | 1.077 |
| 15 | 15 | 14 | 24 | 0 | 1.000 | 1 | 1.071 |
| 25 | 23 | 18 | 28 | 2 | 1.087 | 7 | 1.389 |

Table 18: Lower bound tightness for sparse matrices, $n = 4$, $d = 3$, average sampled basis size $B^* = 20$.

| $B^*$ | $\mathbf{LB}_v$ | $\mathbf{LB}_c$ | $\left|\frac{1}{2}N(p)\right|$ | $\Delta^{(v)}$ | $\rho^{(v)}$ | $\Delta^{(c)}$ | $\rho^{(c)}$ |
|---|---|---|---|---|---|---|---|
| 22 | 13 | 12 | 30 | 9 | 1.692 | 10 | 1.833 |
| 6 | 6 | 4 | 9 | 0 | 1.000 | 2 | 1.500 |
| 13 | 13 | 7 | 14 | 0 | 1.000 | 6 | 1.857 |
| 8 | 8 | 5 | 9 | 0 | 1.000 | 3 | 1.600 |
| 17 | 16 | 9 | 19 | 1 | 1.062 | 8 | 1.889 |
| 20 | 16 | 11 | 22 | 4 | 1.250 | 9 | 1.818 |
| 6 | 6 | 4 | 6 | 0 | 1.000 | 2 | 1.500 |
| 11 | 11 | 6 | 17 | 0 | 1.000 | 5 | 1.833 |
| 19 | 14 | 8 | 24 | 5 | 1.357 | 11 | 2.375 |
| 26 | 15 | 11 | 28 | 11 | 1.733 | 15 | 2.364 |

Table 19: Lower-bound tightness for block diagonal matrices, $n = 6$, $d = 6$, average sampled basis size $B^* = 30$.

| $B^*$ | $\mathbf{LB}_v$ | $\mathbf{LB}_c$ | $\lvert\frac{1}{2}N(p)\rvert$ | $\Delta^{(v)}$ | $\rho^{(v)}$ | $\Delta^{(c)}$ | $\rho^{(c)}$ |
|---|---|---|---|---|---|---|---|
| 40 | 37 | 16 | 89 | 3 | 1.081 | 24 | 2.500 |
| 40 | 38 | 16 | 64 | 2 | 1.053 | 24 | 2.500 |
| 33 | 32 | 14 | 73 | 1 | 1.031 | 19 | 2.357 |
| 39 | 38 | 15 | 59 | 1 | 1.026 | 24 | 2.600 |
| 33 | 33 | 14 | 46 | 0 | 1.000 | 19 | 2.357 |
| 19 | 19 | 10 | 24 | 0 | 1.000 | 9 | 1.900 |
| 21 | 21 | 10 | 28 | 0 | 1.000 | 11 | 2.100 |
| 29 | 29 | 12 | 46 | 0 | 1.000 | 17 | 2.417 |
| 38 | 37 | 15 | 67 | 1 | 1.027 | 23 | 2.533 |
| 22 | 22 | 10 | 27 | 0 | 1.000 | 12 | 2.200 |

Table 20: Lower-bound tightness for low rank matrices, $n = 6$, $d = 6$, average sampled basis size $B^* = 30$.

| $B^*$ | $\mathbf{LB}_v$ | $\mathbf{LB}_c$ | $\lvert\frac{1}{2}N(p)\rvert$ | $\Delta^{(v)}$ | $\rho^{(v)}$ | $\Delta^{(c)}$ | $\rho^{(c)}$ |
|---|---|---|---|---|---|---|---|
| 19 | 19 | 19 | 24 | 0 | 1.000 | 0 | 1.000 |
| 33 | 33 | 33 | 73 | 0 | 1.000 | 0 | 1.000 |
| 40 | 40 | 39 | 89 | 0 | 1.000 | 1 | 1.026 |
| 21 | 21 | 21 | 28 | 0 | 1.000 | 0 | 1.000 |
| 40 | 40 | 39 | 64 | 0 | 1.000 | 1 | 1.026 |
| 23 | 23 | 23 | 33 | 0 | 1.000 | 0 | 1.000 |
| 33 | 33 | 33 | 46 | 0 | 1.000 | 0 | 1.000 |
| 39 | 39 | 38 | 59 | 0 | 1.000 | 1 | 1.026 |
| 22 | 22 | 22 | 27 | 0 | 1.000 | 0 | 1.000 |
| 29 | 29 | 29 | 46 | 0 | 1.000 | 0 | 1.000 |

Table 21: Lower-bound tightness for random matrices, $n = 6$, $d = 6$, average sampled basis size $B^* = 30$.

| $B^*$ | $\mathbf{LB}_v$ | $\mathbf{LB}_c$ | $\lvert\frac{1}{2}N(p)\rvert$ | $\Delta^{(v)}$ | $\rho^{(v)}$ | $\Delta^{(c)}$ | $\rho^{(c)}$ |
|---|---|---|---|---|---|---|---|
| 33 | 33 | 33 | 73 | 0 | 1.000 | 0 | 1.000 |
| 39 | 39 | 38 | 59 | 0 | 1.000 | 1 | 1.026 |
| 27 | 27 | 27 | 42 | 0 | 1.000 | 0 | 1.000 |
| 40 | 40 | 39 | 64 | 0 | 1.000 | 1 | 1.026 |
| 19 | 19 | 19 | 24 | 0 | 1.000 | 0 | 1.000 |
| 29 | 29 | 29 | 46 | 0 | 1.000 | 0 | 1.000 |
| 33 | 33 | 33 | 46 | 0 | 1.000 | 0 | 1.000 |
| 23 | 23 | 23 | 33 | 0 | 1.000 | 0 | 1.000 |
| 38 | 38 | 37 | 67 | 0 | 1.000 | 1 | 1.027 |
| 21 | 21 | 21 | 28 | 0 | 1.000 | 0 | 1.000 |

Table 22: Lower-bound tightness for sparse matrices, $n = 6$, $d = 6$, average sampled basis size $B^* = 30$.

| $B^*$ | $LB_v$ | $LB_c$ | $\lvert\frac{1}{2}N(p)\rvert$ | $\Delta^{(v)}$ | $\rho^{(v)}$ | $\Delta^{(c)}$ | $\rho^{(c)}$ |
|---|---|---|---|---|---|---|---|
| 32 | 32 | 14 | 50 | 0 | 1.000 | 18 | 2.286 |
| 32 | 32 | 18 | 69 | 0 | 1.000 | 14 | 1.778 |
| 25 | 25 | 11 | 39 | 0 | 1.000 | 14 | 2.273 |
| 17 | 17 | 10 | 22 | 0 | 1.000 | 7 | 1.700 |
| 24 | 24 | 12 | 33 | 0 | 1.000 | 12 | 2.000 |
| 32 | 32 | 18 | 71 | 0 | 1.000 | 14 | 1.778 |
| 35 | 35 | 18 | 52 | 0 | 1.000 | 17 | 1.944 |
| 36 | 35 | 17 | 63 | 1 | 1.029 | 19 | 2.118 |
| 20 | 20 | 11 | 23 | 0 | 1.000 | 9 | 1.818 |
| 24 | 24 | 12 | 38 | 0 | 1.000 | 12 | 2.000 |

### E.7 ADDITIONAL OUT-OF-DISTRIBUTION RESULTS

We evaluate a Transformer trained on polynomials with degree 12 and average basis size 30 and full-rank dense matrices on the OOD Grid, i.e., 6 variables, degrees $\{10, 12, 14\}$, average basis sizes $\{20, 25, 35, 40\}$, matrix structures $\{$sparse, low-rank, block diagonal, full-rank dense$\}$.

We report the share of instances that are feasible after repair and scored expansion as well as the average number of expansions performed.

Table 23: Out-of-distribution results for sparse matrices. Transformer trained on full-rank dense matrices.

| Degree | Variables | Monomials | Feasible after repair | Feasible after scored expansion | Expansions |
|---|---|---|---|---|---|
| 10 | 6 | 20 | 0.17 | 0.37 | 0.83 |
| 10 | 6 | 25 | 0.11 | 0.23 | 0.89 |
| 10 | 6 | 35 | 0.06 | 0.22 | 0.94 |
| 10 | 6 | 40 | 0.06 | 0.15 | 0.94 |
| 12 | 6 | 20 | 0.26 | 0.33 | 0.74 |
| 12 | 6 | 25 | 0.17 | 0.21 | 0.83 |
| 12 | 6 | 35 | 0.13 | 0.20 | 0.87 |
| 12 | 6 | 40 | 0.11 | 0.13 | 0.89 |
| 14 | 6 | 20 | 0.22 | 0.32 | 0.78 |
| 14 | 6 | 25 | 0.20 | 0.24 | 0.80 |
| 14 | 6 | 35 | 0.13 | 0.22 | 0.87 |
| 14 | 6 | 40 | 0.10 | 0.14 | 0.90 |
| 16 | 6 | 20 | 0.13 | 0.30 | 0.87 |
| 16 | 6 | 25 | 0.14 | 0.30 | 0.86 |
| 16 | 6 | 35 | 0.16 | 0.14 | 0.84 |
| 16 | 6 | 40 | 0.06 | 0.23 | 0.94 |

Table 24: Out-of-distribution results for full-rank dense matrices. Transformer trained on full-rank dense matrices.

| Degree | Variables | Monomials | Feasible after repair | Feasible after scored expansion | Expansions |
|---|---|---|---|---|---|
| 10 | 6 | 20 | 1.00 | 0.00 | 0.00 |
| 10 | 6 | 25 | 1.00 | 0.00 | 0.00 |
| 10 | 6 | 35 | 0.98 | 0.02 | 0.02 |
| 10 | 6 | 40 | 0.89 | 0.06 | 0.11 |
| 12 | 6 | 20 | 1.00 | 0.00 | 0.00 |
| 12 | 6 | 25 | 1.00 | 0.00 | 0.00 |
| 12 | 6 | 30 | 1.00 | 0.00 | 0.00 |
| 12 | 6 | 35 | 1.00 | 0.00 | 0.00 |
| 12 | 6 | 40 | 0.82 | 0.16 | 0.18 |
| 14 | 6 | 20 | 1.00 | 0.00 | 0.00 |
| 14 | 6 | 25 | 1.00 | 0.00 | 0.00 |
| 14 | 6 | 35 | 0.93 | 0.07 | 0.07 |
| 14 | 6 | 40 | 0.77 | 0.19 | 0.23 |
| 16 | 6 | 20 | 1.00 | 0.00 | 0.00 |
| 16 | 6 | 25 | 1.00 | 0.00 | 0.00 |
| 16 | 6 | 35 | 0.86 | 0.14 | 0.14 |
| 16 | 6 | 40 | 0.66 | 0.28 | 0.35 |

Table 25: Out-of-distribution results for block-diagonal matrices. Transformer trained on full-rank dense matrices.

| Degree | Variables | Monomials | Feasible after repair | Feasible after scored expansion | Expansions |
|---|---|---|---|---|---|
| 10 | 6 | 20 | 0.18 | 0.27 | 0.82 |
| 10 | 6 | 25 | 0.06 | 0.16 | 0.94 |
| 10 | 6 | 35 | 0.01 | 0.08 | 0.99 |
| 10 | 6 | 40 | 0.00 | 0.03 | 1.01 |
| 12 | 6 | 20 | 0.17 | 0.20 | 0.83 |
| 12 | 6 | 25 | 0.04 | 0.14 | 0.96 |
| 12 | 6 | 35 | 0.00 | 0.06 | 1.00 |
| 12 | 6 | 40 | 0.00 | 0.00 | 1.00 |
| 14 | 6 | 20 | 0.18 | 0.25 | 0.82 |
| 14 | 6 | 25 | 0.08 | 0.15 | 0.92 |
| 14 | 6 | 35 | 0.00 | 0.05 | 1.00 |
| 14 | 6 | 40 | 0.00 | 0.05 | 1.00 |
| 16 | 6 | 20 | 0.09 | 0.30 | 0.91 |
| 16 | 6 | 25 | 0.02 | 0.20 | 0.98 |
| 16 | 6 | 35 | 0.00 | 0.05 | 1.00 |
| 16 | 6 | 40 | 0.00 | 0.03 | 1.00 |

While the Transformer generalizes well on instances with similar structure to the training data (i.e. other full-rank dense matrices), it struggles with instances with different structure. We compare this with a Transformer trained on polynomials with degree 12 and average basis size 30 and sparse matrices on the same OOD Grid.

The results are shown in Table 26, Table 27, and Table 28. The Transformer trained on sparse matrices generalizes well to other structures. We conjecture that this is because the Transformer is forced to learn the structure of the sparse matrices, which in particular misses many cross-terms, i.e., products of two distinct monomials from the basis.

Table 26: Out-of-distribution results for sparse matrices, Transformer trained on sparse matrices.

| Degree | Variables | Monomials | Feasible after repair | Feasible after scored expansion | Expansions |
|--------|-----------|-----------|-----------------------|----------------------------------|------------|
| 10 | 6 | 20 | 0.98 | 0.00 | 0.02 |
| 10 | 6 | 25 | 0.90 | 0.02 | 0.10 |
| 10 | 6 | 35 | 0.92 | 0.01 | 0.08 |
| 10 | 6 | 40 | 0.86 | 0.06 | 0.14 |
| 12 | 6 | 20 | 0.90 | 0.01 | 0.10 |
| 12 | 6 | 25 | 0.90 | 0.01 | 0.10 |
| 12 | 6 | 35 | 0.89 | 0.03 | 0.11 |
| 12 | 6 | 40 | 0.88 | 0.05 | 0.12 |
| 14 | 6 | 20 | 0.93 | 0.04 | 0.07 |
| 14 | 6 | 25 | 0.90 | 0.02 | 0.10 |
| 14 | 6 | 35 | 0.81 | 0.13 | 0.19 |
| 14 | 6 | 40 | 0.80 | 0.08 | 0.20 |
| 16 | 6 | 20 | 0.73 | 0.18 | 0.27 |
| 16 | 6 | 25 | 0.76 | 0.21 | 0.24 |
| 16 | 6 | 35 | 0.67 | 0.16 | 0.33 |
| 16 | 6 | 40 | 0.55 | 0.32 | 0.45 |

Table 27: Out-of-distribution results for full-rank dense matrices, Transformer trained on sparse matrices.

| Degree | Variables | Monomials | Feasible after repair | Feasible after scored expansion | Expansions |
|--------|-----------|-----------|-----------------------|----------------------------------|------------|
| 10 | 6 | 20 | 1.00 | 0.00 | 0.00 |
| 10 | 6 | 25 | 1.00 | 0.00 | 0.00 |
| 10 | 6 | 35 | 1.00 | 0.00 | 0.00 |
| 10 | 6 | 40 | 1.00 | 0.00 | 0.00 |
| 12 | 6 | 20 | 1.00 | 0.00 | 0.00 |
| 12 | 6 | 25 | 1.00 | 0.00 | 0.00 |
| 12 | 6 | 30 | 1.00 | 0.00 | 0.00 |
| 12 | 6 | 35 | 1.00 | 0.00 | 0.00 |
| 12 | 6 | 40 | 0.99 | 0.01 | 0.01 |
| 14 | 6 | 20 | 1.00 | 0.00 | 0.00 |
| 14 | 6 | 25 | 1.00 | 0.00 | 0.00 |
| 14 | 6 | 35 | 0.91 | 0.09 | 0.09 |
| 14 | 6 | 40 | 0.78 | 0.20 | 0.22 |
| 16 | 6 | 20 | 1.00 | 0.00 | 0.00 |
| 16 | 6 | 25 | 0.99 | 0.01 | 0.01 |
| 16 | 6 | 35 | 0.71 | 0.25 | 0.29 |
| 16 | 6 | 40 | 0.55 | 0.33 | 0.45 |

Table 28: Out-of-distribution results for blockdiagonal matrices, Transformer trained on sparse matrices.

| Degree | Variables | Monomials | Feasible after repair | Feasible after scored expansion | Expansions |
|--------|-----------|-----------|-----------------------|---------------------------------|------------|
| 10 | 6 | 20 | 0.99 | 0.00 | 0.01 |
| 10 | 6 | 25 | 0.97 | 0.03 | 0.03 |
| 10 | 6 | 35 | 0.97 | 0.03 | 0.03 |
| 10 | 6 | 40 | 0.89 | 0.11 | 0.11 |
| 12 | 6 | 20 | 1.00 | 0.00 | 0.00 |
| 12 | 6 | 25 | 0.98 | 0.01 | 0.02 |
| 12 | 6 | 35 | 0.95 | 0.04 | 0.05 |
| 12 | 6 | 40 | 0.87 | 0.12 | 0.13 |
| 14 | 6 | 20 | 0.98 | 0.02 | 0.02 |
| 14 | 6 | 25 | 0.97 | 0.03 | 0.03 |
| 14 | 6 | 35 | 0.85 | 0.13 | 0.15 |
| 14 | 6 | 40 | 0.77 | 0.22 | 0.23 |
| 16 | 6 | 20 | 0.82 | 0.17 | 0.18 |
| 16 | 6 | 25 | 0.76 | 0.23 | 0.24 |
| 16 | 6 | 35 | 0.54 | 0.42 | 0.46 |
| 16 | 6 | 40 | 0.45 | 0.51 | 0.55 |

### E.8   ABLATION: RANDOM INITIAL BASIS

To assess the necessity of a meaningful initial basis, we compare our method against two uninformed baselines: (i) a basis consisting only of the constant monomial 1, and (ii) a uniformly sampled random basis from the half Newton polytope. For both baselines, we apply the same greedy repair mechanism used in our full pipeline. We evaluate the repair success rate on three representative configurations:

- Config 1 is the small instance with $n = 4$, $d = 6$, $m = 20$ and dense structure;

- Config 2 is the medium instance with $n = 8$, $d = 20$, $m = 30$ and sparse structure (both appear in Table 1);

- Config 3 is the large-scale $n = 100$, $d = 10$, $m = 20$ block-diagonal instance from Table 2.

Table 29: Greedy repair success, average repaired basis size, and runtime relative to the Newton–polytope baseline for uninformed initial bases.

| Config | Init. | Success | Overhead vs Newton |
|--------|-------|---------|--------------------|
| Config 1 | Const. {1} | 5% | 3.93× |
|          | Random | 4% | 3.54× |
| Config 2 | Const. {1} | 5% | 2.05× |
|          | Random | 5% | 1.96× |
| Config 3 | Const. {1} | 0% | - |
|          | Random | 0% | - |

**Conclusion:** Across all configurations, uninformed initial bases—whether the constant monomial or a randomly sampled subset—lead to extremely low repair success (0–5%) and incur substantial runtime overheads of up to 3×–4× the Newton–polytope method. In particular, repair has no structural signal to exploit, causing the algorithm to revert almost entirely to full Newton bases. These results demonstrate that greedy repair *cannot* compensate for an uninformative starting point; a meaningful, learned initializer is therefore essential for obtaining any reduction in basis size or runtime.

### E.9 ABLATION: DIFFERENT ARCHITECTURES

To isolate the effect of sequence model architecture, we directly compared a Transformer, an LSTM (Hochreiter & Schmidhuber, 1997), and a Mamba model (Gu & Dao, 2023; Dao & Gu, 2024) under a fixed wall-clock training budget of 5 hours on the large benchmark dataset $n = 6$, $d = 10$, $m = 60$, dense structure (cf. Table 2).

- **Transformer:** $d_{\text{model}} = 512$, encoder layers 6, decoder layers 6, 8 attention heads.
  Total parameters: 106,486,272 (106.49M)

- **LSTM:** $d_{\text{model}} = 512$, 8 layers, bidirectional, dropout 0.1.
  Total parameters: 74,227,432 (74.23M)

- **Mamba:** $d_{\text{model}} = 512$, 8 layers, $d_{\text{state}} = 16$, $d_{\text{conv}} = 4$, expand factor 3, unidirectional, cross-attention enabled, dropout 0.1.
  Total parameters: 115,144,168 (115.14M)

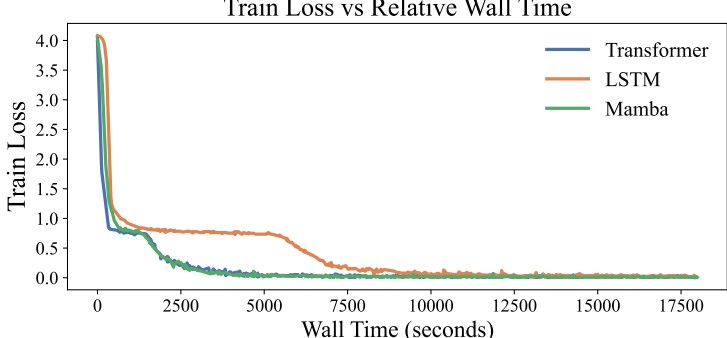

Figure 12: Train loss as a function of relative wall time for different sequence-model architectures under a fixed 5-hour training budget.

| Model | LSTM | Mamba | Transformer |
|---|---|---|---|
| Test classification error | 0.067 | 0.041 | 0.038 |

Table 30: Final test classification error for basis prediction after 5 hours of training.

**Conclusion:** Under a matched training budget and comparable parameter scales, the Transformer and Mamba models exhibit very similar behavior: both converge rapidly and reach substantially lower error than the LSTM baseline. The Transformer achieves the lowest final classification error, but the margin over Mamba is small (0.038 vs. 0.041), and their learning curves track each other closely throughout training. In contrast, the LSTM optimizes more slowly and remains noticeably less accurate despite using a comparable hidden size. Overall, these results suggest that architectures capable of capturing long-range interactions—whether via self-attention or selective state-space mechanisms—are particularly effective for SOS basis prediction.

### E.10 ABLATION: CHECKPOINT PERFORMANCE ACROSS TRAINING

To evaluate how the method's performance evolves throughout training, we assess our Transformer-model at various checkpoints corresponding to different numbers of training samples $m \in \{10000, 100000, 200000, 300000, 400000, 500000, 1000000\}$. We report results for two representative configurations from Table 1: (i) $n = 8$, sparse structure, degree 20, average basis size 30; and (ii) $n = 4$, dense structure, degree 6, average basis size 20. For each, we report test classification error and coverage support and solver runtime.

| samples | error | support coverage | solve time |
|---|---|---|---|
| 10,000 | 0.4955 | 0.00 | 12.59 |
| 100,000 | 0.1583 | 0.04 | 9.45 |
| 200,000 | 0.0382 | 0.11 | 2.59 |
| 300,000 | 0.0159 | 0.34 | 1.77 |
| 400,000 | 0.0101 | 0.56 | 1.05 |
| 500,000 | 0.0067 | 0.60 | 0.93 |
| 1,000,000 | 0.0029 | 0.62 | 0.62 |

Table 31: Checkpoint performance on configuration (i) ($n = 8$, degree 20, average basis size 30).

| samples | error | support coverage | solve time |
|---|---|---|---|
| 10,000 | 0.03003 | 0.17 | 1.13 |
| 100,000 | 0.00095 | 0.67 | 0.67 |
| 200,000 | 0.00078 | 0.71 | 0.43 |
| 300,000 | 0.00065 | 0.71 | 0.40 |
| 400,000 | 0.00063 | 0.71 | 0.40 |
| 500,000 | 0.00061 | 0.71 | 0.40 |
| 1,000,000 | 0.00061 | 0.71 | 0.40 |

Table 32: Checkpoint performance on configuration (ii) ($n = 4$, degree 6, average basis size 20).

**Conclusion:** Across both experiments, performance improves consistently as the number of training samples increases. At very small checkpoints (e.g., $m = 10,000$), the model exhibits high variance and comparatively poor error, reflecting underconstrained estimation. Once $m$ reaches the mid-range (200,000–300,000), error drops by more than an order of magnitude and coverage support stabilizes near zero, indicating that the classifier has effectively learned the underlying structure. Increasing $m$ further provides diminishing but still measurable gains: the lowest errors consistently occur at or near $m = 1,000,000$. Overall, these results show that the method benefits substantially from additional training samples early on, but transitions to a regime of slow incremental improvement once sufficient data is available.

### E.11 CASE STUDY: VAN DER POL OSCILLATOR

We consider the standard Van der Pol system

$$\frac{d^2x}{dt^2} - \mu(1 - x^2)\frac{dx}{dt} + x = 0.$$

As candidate Lyapunov function, we use a polynomial from (Peet & Papachristodoulou, 2012)

$$V(x_1, x_2) = (6.93x_1 - 2.45x_2)^2 + \left(2.45(x_1 + x_1^2 x_2) + 4.48x_2\right)^2$$
$$+ (2.45x_1 - x_2)^2 + (x_1 + x_1^2 x_2 + 1.45x_2)^2.$$

We plug this candidate into `SumOfSquares.jl`. The package automatically constructs a *sufficient* monomial basis for the SOS constraint, namely

$$B_1 = [\, x_2, \ x_1, \ x_1 x_2, \ x_1^2 x_2 \,],$$

which is strictly larger than necessary.

In contrast, the Transformer-model trained on sparse instances with $n = 4$ variables correctly identifies the *minimal* SOS basis

$$B_2 = [\, x_1, \ x_2, \ x_1^2 x_2 \,],$$

which is exactly the set of monomials that appear linearly in the four square terms defining $V$.

### E.12 TRAINING TIME AMORTIZATION

To evaluate the practical trade-off between one-time training cost and per-instance solver speedup, we consider four representative datasets spanning the regimes in Tables 1 and 2:

- **Small:** dense 6-variable, degree-12 instances (cf. Table 1);
- **Medium:** 6 variables, degree 20 (cf. Table 2);
- **Large:** 8 variables, degree 20 (cf. Table 2);
- **XL:** 100 variables, degree 10 (cf. Table 2).

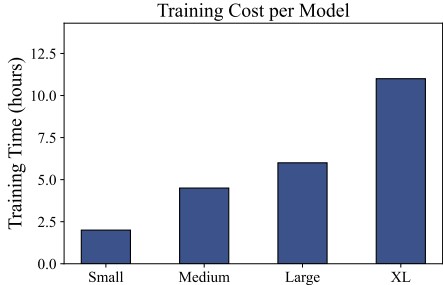 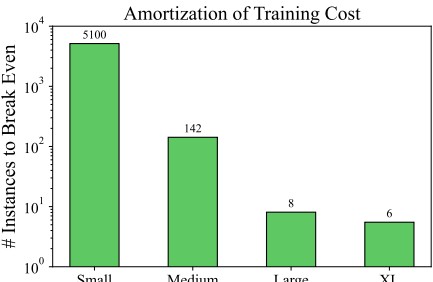

Figure 13: Training cost vs. amortized speedups. Left: total training time (hours) for the four representative problem scales. Right: number of instances required to amortize training, computed as $N_{\text{break-even}} = T_{\text{train}}/\Delta T$, where $\Delta T$ is the average per-instance time saved compared to the best baseline solver (up to the 2-hour timeout for XL). For small instances, training would only amortize after thousands of solves; for large and XL configurations, amortization occurs after roughly 6–8 problems, each saving tens of minutes to about two hours of solver time.

As shown in Figure 13, small-scale models train quickly ($\sim$2 h) but offer only sub-second savings per instance, making amortization impractical unless thousands of solves are required. In contrast, for larger configurations—where a single solve can save from tens of minutes up to the 2-hour timeout—the one-time training cost (4–11 h) is amortized after only 6–8 SOS problems. This demonstrates that learning-augmented basis prediction is most beneficial precisely in the high-dimensional regime where classical SOS methods struggle computationally.

### E.13 QUALITATIVE MODEL BEHAVIOR

We briefly analyze the Transformer's predictions. In simple settings, it learns intuitive algebraic patterns—for instance, including $x_i^2$ whenever higher-order terms like $x_i^4$ appear, echoing common SOS structures. Even in complex cases such as the high-degree experiments in Table 2 (degree 40), the model selects compact bases that outperform naïve degree-based heuristics. Overall, the Transformer captures meaningful dependencies among monomials despite the typical difficulty of fully interpreting its internal representations.

## F PROOFS

**Lemma 2.** *Let $p(\mathbf{x}) \in \mathbb{R}[x_1, \ldots, x_n]$ be a polynomial and $B \subseteq \mathcal{M}_d$ be a basis. Then, the support of $p$ is contained in the set of all pairwise products of monomials in $B$, formally $S(p) \subseteq B \cdot B$ where $B \cdot B = \{m_i \cdot m_j : m_i, m_j \in B\}$. We say that $B$ covers $p(\mathbf{x})$ if $S(p) \subseteq B \cdot B$.*

*Proof.* Let $p \in \mathbb{R}[x_1, \ldots, x_n]$ be a polynomial and $B \subseteq \mathcal{M}_d$ be a basis. If $p$ has an SOS decomposition using basis $B$, then

$$p(\mathbf{x}) = \mathbf{z}_B(\mathbf{x})^\top Q \, \mathbf{z}_B(\mathbf{x})$$

for some PSD matrix $Q$. Let $u \in S(p)$ be any monomial appearing in $p$. Since $p(\mathbf{x}) = \mathbf{z}_B(\mathbf{x})^\top Q \, \mathbf{z}_B(\mathbf{x})$, the monomial $u$ must arise from some product $z_i(\mathbf{x}) z_j(\mathbf{x})$ where $z_i, z_j \in B$. Therefore, $u \in B \cdot B$. $\qquad\square$

**Lemma 5** (Consistency). *Let $B_{cov}$ be a feasible SOS basis obtained by Algorithm 1. If $\eta = |B_{cov}|$ and $|B_{cov}| = |B^*|$, then Algorithm 1 finds a SOS certificate by solving only a single SDP with the smallest possible basis size.*

*Proof.* This is immediate from the definition of $\eta$ and the fact that $|B_{cov}| = |B^*|$. Therefore, the algorithm will find the certificate on the first SDP solve. $\square$

**Lemma 6** (Robustness). *The combined computational cost of Algorithm 1 and Algorithm 2 is at most $k$ times the cost of the standard Newton polytope approach.*

*Proof.* In the worst case, our algorithm fails to find a feasible SOS decomposition until the final attempt with basis size $m_k = N$ (the full sufficient set). By our assumption that the cost of the SDP with basis size $m$ scales as $\Theta(m^\omega)$, the total computational cost is bounded by

$$\Theta\left(\sum_{j=1}^{k} m_j^\omega\right).$$

The cost of the standard Newton polytope approach scales as $\Theta(N^\omega)$.

Therefore, the competitive ratio is bounded by

$$\beta = \frac{\Theta\left(\sum_{j=1}^{k} m_j^\omega\right)}{\Theta(N^\omega)} = \Theta\left(\sum_{j=1}^{k} \frac{m_j^\omega}{N^\omega}\right) = O(k).$$

$\square$

**Lemma 4** (Geometric Expansion). *Let $B_{cov}$ be a coverage-repaired SOS basis obtained by Algorithm 1. Let the coverage rank be $\eta$, set $m_1 = |B_{cov}|$ and choose $\rho > 1$. Then, Algorithm 2 performs at most $1 + \lceil \log_\rho(\eta/m_1) \rceil$ SDP solves and has total cost at most $\Theta\left(\frac{\rho^\omega}{1-\rho^{-\omega}} \eta^\omega\right)$.*

*Proof.* Let the schedule satisfy $m_1 = |B_{cov}|$ and $m_{i+1} = \lceil \rho m_i \rceil$ with $\rho > 1$. By induction, $m_t \geq \rho^{t-1} m_1$. We must reach a basis of size at least the coverage rank $\eta$, so the smallest $j$ with $\rho^{j-1} m_1 \geq \eta$ satisfies $j \leq 1 + \lceil \log_\rho(\eta/m_1) \rceil$.

For the cost bound, note that the cost of our schedule is $\Theta\left(\sum_{t=1}^{j} m_t^\omega\right)$. Further, note that $m_{j-1} < \eta$ and thus $m_j = \lceil \rho m_{j-1} \rceil \leq \rho\eta$. For any $t \leq j$, $m_t \leq \frac{m_j}{\rho^{j-t}} \leq \frac{\rho\eta}{\rho^{j-t}}$. Therefore, $\sum_{t=1}^{j} m_t^\omega \leq (\rho\eta)^\omega \sum_{s=0}^{j-1} \rho^{-\omega s} \leq \frac{\rho^\omega}{1-\rho^{-\omega}} \eta^\omega$, which yields the claimed $\Theta\left(\frac{\rho^\omega}{1-\rho^{-\omega}} \eta^\omega\right)$. $\square$

Finding minimal SOS bases is computationally hard as it involves $\ell_0$ minimization. Instead, we establish lower bounds on the size of a basis that are often tight in practice.

**Lemma 7** (Combinatorial bound). *For any SOS basis $B$ of polynomial $p(\mathbf{x})$ with support $S(p)$, we have $|B| \geq \frac{\sqrt{1+8|S(p)|}-1}{2}$.*

*Proof.* Suppose we are given a valid SOS decomposition $p(\mathbf{x}) = \mathbf{z}_B(\mathbf{x})^\top Q \, \mathbf{z}_B(\mathbf{x})$ of $p$. Then, we can rewrite $\mathbf{z}_B(\mathbf{x})^\top Q \, \mathbf{z}_B(\mathbf{x})$ as a sum of squares of linear forms:

$$\mathbf{z}_B(\mathbf{x})^\top Q \, \mathbf{z}_B(\mathbf{x}) = \sum_{i,j} Q_{ij} z_i(\mathbf{x}) z_j(\mathbf{x}).$$

Therefore, the number of distinct monomials in $\mathbf{z}_B(\mathbf{x})^\top Q \, \mathbf{z}_B(\mathbf{x})$ is at most $\sum_{k=1}^{|B|} k = \frac{|B|(|B|+1)}{2}$. Since $\mathbf{z}_B(\mathbf{x})^\top Q \, \mathbf{z}_B(\mathbf{x}) = p(\mathbf{x})$ we have that $|S(p)| \leq \frac{|B|(|B|+1)}{2}$. Therefore, $|B| \geq \frac{\sqrt{1+8|S(p)|}-1}{2}$. $\square$

Intuitively, the combinatorial bound follows from the fact that a basis of $m$ monomials can generate at most $\frac{m(m+1)}{2}$ distinct monomials. Finally, the Newton polytope vertices lemma shows that certain vertices of the Newton polytope must appear in any SOS basis.

**Lemma 8** (Newton polytope vertices). *For any SOS basis $B$ of polynomial $p(\mathbf{x})$ with support $S(p)$ and half Newton polytope vertices $V = \text{vertices}(\frac{1}{2}N(p))$, we have $\{u \in V : u^2 \in S(p)\} \subseteq B$.*

*Proof.* Suppose $u \in V$ is a vertex of $\frac{1}{2}N(p)$ and $u^2 \in S(p)$. Further, we know that due to $u$ being a vertex of $\frac{1}{2}N(p)$, there are no other monomials $u_1, u_2 \in \frac{1}{2}N(p)$ such that $u_1 u_2 = u^2$. Therefore, the only way to obtain $u^2$ in any SOS decomposition of $p$ is to include $u$ in the basis. Since this holds for all vertices of $\frac{1}{2}N(p)$ for which we have $u^2 \in S(p)$, we have that any SOS decomposition of $p$ must include all monomials in $V$. $\square$

We now study schedule selection with an *unknown* coverage rank. Given an initial basis size $m_1$ and a geometric expansion factor $\rho > 1$, let $j$ denote the first iteration where Algorithm 2 finds a feasible solution with basis size $m_j$. Feasibility requires

$$\rho^j m_1 \geq m_j \geq \eta.$$

In practice, we do not know the true coverage rank $\eta$ in advance. If $\eta$ were known, we would jump to size $\eta$ in a single step. Since $\eta$ is instance-dependent and unknown at expansion time, we model

$$X = \eta$$

as a random variable and pose the stochastic optimization problem

$$\min_{\rho > 1} \quad \sum_{s=1}^{k(\rho)} (\rho^s m_1)^\omega$$

$$\text{s.t.} \quad k(\rho) = \left\lceil \log_\rho \left( \frac{\eta}{m_1} \right) \right\rceil,$$

$$\rho > 1.$$

Using empirical risk minimization, let $X_1, \ldots, X_M$ be samples of $X$ (and allow $m_{1,i}$ to vary if desired). We then solve

$$\min_{\rho > 1} J(\rho) = \sum_{i=1}^{M} \sum_{s=1}^{k_i(\rho)} (\rho^s m_{1,i})^\omega$$

$$\text{s.t.} \quad k_i(\rho) = \left\lceil \log_\rho \left( \frac{X_i}{m_{1,i}} \right) \right\rceil, \quad \quad \text{(ERM)}$$

$$\rho > 1.$$

Because the cost function is increasing in basis size and $k_i(\rho)$ decreases with $\rho$ (and is piecewise constant), it suffices to evaluate a finite grid of $\rho$ values and select the one with the smallest cost.

**Theorem 9** (Finite candidate grid for optimal $\rho$). *Let $J(\rho)$ denote the ERM objective introduced above. Then, there exists a finite grid of $\rho$ values such that the minimizer of $J(\rho)$ lies in the grid.*

*Proof.* We first show that $J(\rho)$ has a minimizer. Note that $m_{1,i} \leq X_i$ for all $i$, therefore $k_i(\rho) \geq 1$. Since $\{\frac{X_i}{m_{1,i}}\}_{i=1}^{M}$ is bounded, there exists $\rho_{\max} > 1$ such that $k_i(\rho_{\max}) = 1$ for all $i$. Further, for all $\rho > \rho_{\max}$ we have that $J(\rho) \geq J(\rho_{\max})$. Further, we have that as $\rho \to 1$, $J(\rho) \to \infty$. Therefore, $J(\rho)$ has a minimizer that lies in $[\rho_{\min}, \rho_{\max}]$ for some $\rho_{\max} > \rho_{\min} > 1$.

Next, note that all $k_i(\rho)$ are piecewise constant, and there are finitely many breakpoints. Therefore, there exists a finite grid of $\rho$ values $\rho_1, \ldots, \rho_L$ such that for all $(\rho_i, \rho_{i+1})$ all $k_i(\rho)$ are constant. Since $(\rho^s m_{1,i})^\omega$ is strictly increasing in $\rho$, we have that $J(\rho)$ is strictly increasing for all intervals $(\rho_i, \rho_{i+1})$. Therefore, the minimizer of $J(\rho)$ lies in the left endpoint of one of the intervals $(\rho_i, \rho_{i+1})$. $\square$

**Remark 10.** *Further refinements of the proposed approach are possible. Instead of a fixed geometric expansion, any increasing sequence of basis sizes $m_1 < m_2 < \cdots < m_k$ can be used and potentially similarly be optimized via ERM. Here, we chose the geometric expansion for its simplicity.*

