# OpenReview forum: "Neural Sum-of-Squares: Certifying the Nonnegativity of Polynomials with Transformers"
_ICLR.cc/2026/Conference — ICLR 2026 Poster_

### Official Review · Reviewer_JFgt · 2025-10-28

**Soundness:** 3
**Presentation:** 3
**Contribution:** 3
**Rating:** 6
**Confidence:** 3

**Summary:**

This paper presents a novel learning-augmented approach to Sum-of-Squares (SOS) certification, using transformers to predict small monomial bases resulting in large reduction of the corresponding SDP problem. An iterative fallback mechanism is deployed to ensure correctness of the approach. Empirical results demonstrate important speedups, as well as robustness across polynomial structures, supported by experiments on extensive synthetic datasets. The writing is clear, figures are informative, and the method’s motivation, i.e., addressing the core computational bottleneck in SOS programming, is convincing.

**Strengths:**

The approach is novel, sound, supported by experiments, and demonstrates consistent speedups across different polynomial structures. The writing of the main part is mostly clear, and the overall methodology is both coherent and practically relevant.

**Weaknesses:**

While the approach is promising, the evaluation is limited to synthetic data, making its generalization to real or benchmark SOS problems unclear. The theoretical analysis offers mainly worst-case guarantees and does not fully characterize how prediction quality influences solver performance. Comparisons with modern sparsity-based SOS solvers are brief.

### Major Points

- The paper relies on several different transformer architectures for different experiment grids, but the reasons behind these architectural choices are not clearly explained. In addition, training appears to require very large datasets, but there is no information about the training costs. As a result, it is difficult to evaluate the practical trade-off between the cost of learning and the runtime improvements obtained during SOS solving.

- Although the paper claims real-world SOS benchmarks are not widely available, there are well-established applications, such as Lyapunov stability certification in control. Testing even a small selection of such problems would greatly strengthen the practical relevance and demonstrate generalization beyond synthetic data.

- Theoretical results give useful worst-case guarantees for the fallback mechanism, but they do not provide any attempt of a principled explanation of when the transformer should produce accurate predictions that yield substantial speedups. At present, the conditions under which the learning generalizes reliably (e.g., to sparse or block-diagonal structure) are supported only by experiments.


### Minor Points

- It would be appropriate to distinguish between a polynomial $p \in \mathbb{R}[x_1,\dots,x_n]$ and its evaluation $p(\mathbf{x}) \in \mathbb{R}$ at some point $\mathbf{x} \in \mathbb{R}^n$.
- Line 106: “…smallest possible basis *for which a given polynomial* $p$ admits an SOS decomposition.”
- Lines 126–129: rephrase for clarity.
- Figures should be placed and discussed immediately after they are introduced. For example, Figure 1 is only referred to at Line 185. Apply this consistently throughout the paper (**especially Section 4**) and appendix for all figures and tables to improve readability.
- Line 210: there should be an intersection with $\mathbf{Z}^n_{\geq 0}$, as the networks likely do not predict rational exponents.
- A lemma typically denotes a nontrivial intermediate theoretical result. Since Lemma 1 is a direct observation from the SOS definition, calling it a remark may probably be more appropriate.
- It is not necessary to include the Newton polytope in the input to your algorithms, as the polytope depends on the true input $p$.
- Line 343: replace *is* by *are*.
- Figure 3 (or Section 4.1) could be improved by reporting, for example, the average gap between the Transformer’s prediction and the minimal necessary basis size.
- Figure 6 seems more appropriate for the main text, where it could help illustrate the concept on your simple toy example.
- Line 1001 (Table 9): k should be in math mode.
- Use $\hat{B}$ consistently for a single meaning (see Lines 1069–1070).
- Algorithm 2: write the concatenation $(B_{\text{cov}}, C)$ properly. Also, use t\in`\{2,...,k\}`  instead of
$t \in (2, \dots, k)$.
- Line 1511: $|S|$ should be $|S(p)|$.

**Questions:**

## Questions

1. Why is it not reasonable to pick a single transformer architecture that is not dependent on the data, and that has some specialized features, such as equivariance under variable ordering?

2. You generate non-SOS polynomials by perturbing PSD spectra in order to test the overhead of your approach. Did you also consider the inverse procedure — starting from non-SOS polynomials and projecting or adjusting them into the SOS cone — to obtain more realistic and challenging borderline SOS instances for the training data?

3. Any rationale for why your approach underperforms on small and simple test cases?

4. In Table 2, what is the structure of polynomials used for testing? Averages are taken over how many instances?

5. In Figure 7(ii), why is #$\text{SDPs}$ (y-axis) between 0 and 1?

6. What is $B^{*}$ in Table 1?

7. What is the meaning of the numbers above the bars in Figure 3?

8. Why is $L=2$ used in Experiment 3? By “combined approach,” do you mean first using the greedy approach to ensure the necessary condition from Lemma 1, then using permutation-based expansion with $L=2$?

9. What modifications are necessary for your approach to be applied to general (un)constrained polynomial optimization problems?

10. Which repair mechanism was used in Figure 5(ii)?

11. Line 1563: What is $T$?

---

> ### Author Response · Authors · 2025-11-21
> **Clarifications Addressing Evaluation Scope, Training Costs, and Generalization Behavior**
>
> # Answer
>
> We thank the reviewer for the thoughtful and constructive feedback.
>
> # Major Comments
>
> ## Trade-Off Between Model Training and SOS Solving Time
>
> All experiments use a uniform encoder-only transformer architecture, with only minor adjustments (depth/width) to accommodate input size differences across polynomial families.
>
> Dataset generation is fast: even our largest (100 variables) took under 2 hours, all others under 10 minutes. Training on an NVIDIA A40 GPU took 4–5 hours for most models, up to 11 hours for the largest (100 variables). This is offset by large solver speedups—baseline solvers often time out after 2 hours (Table 2), while our method consistently finds certificates due to much smaller SDPs. For small instances, baseline solvers are already fast, so speedups are minor.
>
> ## Evaluation on Non-Synthetic Benchmarks
>
> We agree that real-world evaluation would strengthen the work. However, many SOS applications (e.g., in control) require _synthesizing_ a polynomial  before verification. Since our method assumes a fixed polynomial and focuses on certification, suitable benchmarks are not readily available.
>
> For illustration, we have evaluated our method on the classical van der Pol oscillator using a pre-specified Lyapunov candidate. In this example, our transformer successfully found the minimal basis. We emphasize that this serves as an illustrative case study; results will be incorporated in the revised manuscript.
>
> Our work complements recent efforts such as [1], which use transformers to synthesize Lyapunov candidates but do not address verification or exploit sparsity.
>
> [1] Alfarano et al., Global Lyapunov functions: a long-standing open problem in mathematics, with symbolic transformers, 2024.
>
> ## Theoretical Characterization of Prediction Success
>
> Thank you for this important point. While formal guarantees on when the transformer predicts a minimal basis are currently not feasible in high-dimensional combinatorial settings, our out-of-distribution experiments (Tables 23–28 in the appendix) provide some insight:
>
> Transformers trained on sparse polynomials generalize well, while those trained only on dense instances generalize poorly—even on similar data. We conjecture that sparse polynomials, with many missing cross-terms, create more complex and varied combinatorial structures. Training on such data encourages the model to learn richer, more general representations—matching our empirical results.
>
> # Questions
> 	1.	Why not use a single, architecture-agnostic transformer?
> A universal Transformer is an appealing idea. Naturally, Transformers trained on $n$ variables can also handle polynomials with less than $n$ variables. Note however, that we explicitly exploit that the Transformer is not equivariant with respect to variable ordering as this allows for the generating signals for the scoring mechanism.
>
> 	2.	Did you consider projecting non-SOS polynomials into the SOS cone to generate challenging borderline examples?
> Thank you for this suggestion. While we did not use this in our main experiments, this is an interesting direction for generating harder training data.
>
> 	3.	Why underperformance on small and simple test cases?
> The transformer is in fact highly accurate on small cases (see Fig. 4(i)), but the overhead of inference eliminates any runtime gain in this regime.
>
> 	4.	What is the structure of polynomials in Table 2, and how many instances were averaged?
>  For 100-variable cases, we used block-diagonal structures; for the n=6, d=40 case, dense matrices; and sparse matrices for intermediate sizes (with average basis sizes around 60); averaged over 10 instances (cf. Appendix D for dataset details)
>
> 	5.	#SDPs between 0 and 1 in Fig. 7(ii)?
> The plot shows the average number of additional SDP solves needed beyond the initial solve. On average, it rarely exceeded one.
>
> 	6.	$B^*$ in Table 1?
> $B^*$ denotes the average basis size used to generate the polynomial instances.
>
> 	7.	Figure 3?
> These indicate total solver time (corresponding to the dense baseline in Table 1). We will clarify this in the text.
>
> 	8.	Experiment 3 clarification?
> Yes — the combined method first applies the greedy repair followed by permutation-based expansion with L = 2. We will clarify this.
>
> 	9.	(un)constrained polynomial optimization?
> One approach is to use a sampling-based method or golden section search to estimate a lower bound $\lambda$ and verify $p - \lambda$ via our pipeline. Alternatively, one could train models to directly predict minimal bases for $p(x) - \lambda$, given suitable data.
>
> 	10.	Fig. 5(ii)?
> This plot uses the combined repair mechanism mentioned above (greedy + L = 2 permutation expansion).
>
> 	11.	T in Line 1563?
> This is a typo; T denotes the time to solve an SDP of given size.
>
>
>
> # Minor Points
>
> Thank you for your close reading. We will fix all minor issues as suggested.
>
>
> Thank you again for your helpful feedback. We hope these clarifications address all concerns.

---

> > ### Comment · Reviewer_JFgt · 2025-11-24
> > **First response to the authors' rebuttal**
> >
> > Dear Authors,
> >
> > Thank you for your efforts in clarifying several of the raised points. However, the currently available version of the manuscript appears to be *identical* to the initial submission.
> >
> > Would it be possible for you to provide the revised and improved version, with the major changes clearly highlighted (in blue)? This would allow me to better assess the quality of your responses and to reevaluate the strength of the submission. I believe the other Reviewers would also find this helpful.
> >
> > Thank you in advance.

---

> > > ### Author Response · Authors · 2025-11-26
> > > **Upload manuscript**
> > >
> > > We have now uploaded the revised manuscript, thank you again for your feedback. We have also added an official comment to all reviewers highlighting the updates in the revision, including the changes addressing your concerns. Please let us know whether further clarification is needed.

---

### Official Review · Reviewer_wmbZ · 2025-11-01

**Soundness:** 3
**Presentation:** 3
**Contribution:** 3
**Rating:** 6
**Confidence:** 4

**Summary:**

This paper proposes a learning-augmented algorithm for certifying the nonnegativity of polynomials via the sum-of-squares (SOS) criterion, a classic computational bottleneck in polynomial optimization. The authors leverage a Transformer model to predict a compact monomial basis for a polynomial, reducing the size (and cost) of the associated semidefinite program (SDP) used for SOS verification. They propose a fallback mechanism to ensure correctness for incomplete predictions, provide theoretical analysis on algorithmic efficiency, and empirically validate the method over a very large synthetic dataset and multiple benchmark scenarios, demonstrating substantial speedup over state-of-the-art methods while maintaining correctness guarantees.

**Strengths:**

1. The paper tackles the persistent computational challenge of SOS certification with a hybrid approach that smartly decomposes basis selection into a learning-driven task, while retaining the robustness of fallback combinatorial methods. The integration of a Transformer model into this traditionally symbolic and combinatorial problem is notably creative.

2. The systematic repair and iterative expansion procedure ensures that, even when the learned model predictions are imperfect, the algorithm retains the worst-case correctness guarantees of classical approaches. The underlying fallback logic is clearly explained and theoretically justified.

3. The empirical evaluation, including Table 1 and Table 2, is broad, comparing against strong baselines such as Newton polytope and full-basis SDP, and modern solvers like SoS.jl and TSSOS. The results are benchmarked across polynomial structure (dense, sparse, low-rank, block-diagonal), variable and degree diversity, and stress test the approach on scalability and distribution shifts.

4. The paper rigorously analyzes the robustness, computational savings, and expansion scheduling, connecting prediction quality to overall efficiency and validating these claims empirically.

5. The paper delivers detailed ablations on design choices such as permutation count and expansion scheduling, showing how performance and computational tradeoffs play out in practice.

**Weaknesses:**

1. The empirical results rely almost entirely on synthetically generated polynomial instances. The lack of real-world or domain-specific SOS instances is acknowledged as a limitation, but is still significant. This limits the evidence for generalization beyond the synthetic distribution and may mask domain-specific failings.

2. Although Table 1 and other results show strong average-case speedups, there is only brief mention of numerical instability (e.g., SCS used for low-rank cases due to issues with MOSEK). There is little detailed discussion on how the approach fares with poorly conditioned or pathological instances, or the frequency and severity of failed or unreliable SDP solves.

3. While there are useful heatmaps and success rate statistics, the paper stops short of a rigorous exploration of the behavioral regime where the Transformer model fails most catastrophically (e.g., in dense and low-rank polynomials). Further, it remains unclear whether model “hallucinations” or systematic biases exist in certain polynomial structures that could lead to silent failure, even after repair.

4. The algorithm for fallback/expansion and scoring monomials by permutation frequency is mathematically described, but several technical implementation choices and their rationale (e.g., why a specific permutation count $L=4\sim8$ balances computational cost and recall) are only empirically motivated. It would be beneficial to see more theoretical discussion or even bounds on the expected number of necessary expansions under mild error assumptions. Furthermore, the text can be unclear on the relationship between minimum basis selection and coverage repair, especially when the predicted basis is far from minimal.

5. The paper claims generalization to degrees and distributions not seen in training, but the OOD analysis (while a nice touch) remains somewhat superficial and is itself conducted over synthetic shifts. More rigorous OOD characterization (such as sharp ablation on unfamiliar sparsity, more adversarial polynomials, or empirically ill-posed SDPs) is missing.

6. The model is said to predict “near-minimal” bases, but there is no formal guarantee, and the possibility of pathological overextension (and thus loss of computational gains) is not fully explored in the main paper.

7. The repair algorithms are shown to work well overall, but the impact of their greedy design is not fully characterized in failure cases, such as cases with many missing basis elements or in noisy/perturbed input. Table 9 and Table 10 provide some view on combinatoric recovery, but the exhaustive performance of the repair mechanism, especially as the dataset difficulty increases, is left for future work.

**Questions:**

1. Are there plans or possibilities to evaluate the framework on real-world, domain-specific SOS problems (e.g., systems or control, chemical design, robotics)? Can the authors comment on what barriers exist to acquiring such benchmarks, or how the method would fare on “messier” non-synthetic input?

2. Could the authors expand on the types and frequency of solver instabilities encountered, e.g., when switching from MOSEK to SCS? Are there specific structures or matrix conditions that systematically cause difficulties? What practical guidelines could be recommended?

3. For the few cases where the coverage-repaired basis remains insufficient and a large number of basis expansions are required, what failure patterns emerge? Do these correlate with any specific polynomial properties, and can they be detected a priori?

4. Is there any pathway to either guarantee (or at least probabilistically certify) that the learned basis is minimal, or offer reliable bounds on non-minimality? Are there pathological cases in the data generation or in-the-wild polynomials that routinely “confuse” the model?

5. For the non-SOS (negative) polynomials, what is the practical cost for fallback expansion, especially for high-degree/large-variable settings? Could this be further accelerated with early termination heuristics, and would this threaten guarantees?

6. Have the authors tried scaling beyond 100 variables or polynomials with very large/structured sparsity—are there new bottlenecks for the Transformer or SDP solvers in such regimes?

7. Are there possible theoretical refinements to the expansion schedule or the permutation scoring to further reduce worst-case number of SDPs or average expansion cost?

**Details Of Ethics Concerns:**

N/A.

---

> ### Author Response · Authors · 2025-11-21
> **Author Rebuttal and Clarifications**
>
> We thank the reviewer and have revised the manuscript to clarify assumptions, failure modes, and solver instability.
> ## External Validity and OOD
>
> **Synthetic vs. real-world evaluation**
> We now include a van der Pol oscillator example as an illustrative real-world test, where our method yields a slightly smaller valid basis than Newton-polytope, though this does not constitute strong real-world evidence.
>
> Broader real-world evaluation is limited by a lack of suitable benchmarks: most SOS domains focus on synthesis, where coefficients and sparsity are variable and no fixed basis exists. Current benchmarks are often small and dense, and even recent Transformer-based Lyapunov work [1] uses synthetic, low-dimensional examples.
>
> [1] Alfarano et al., Global Lyapunov functions: a long-standing open problem in mathematics, with symbolic transformers, 2024.
>
> **OOD generalization**
> Tables 23–28 show that Transformers trained on _sparse_ instances generalize better OOD than those trained on _dense_ data.
> This suggests the model benefits from learning sparse structure; we clarify this point and its limits in the paper.
>
> ---
>
> ## Failure Modes and Model Bias
>
> **Failure modes**
> The Transformer mainly errs by missing monomials on dense or low-rank instances. Even a single omission can cause failure, but since false negatives are typically few, repair and expansion quickly recover a feasible basis with minimal extra SDPs. Thus, speedups are retained despite initial mistakes.
>
> **No silent failures**
> All predicted bases are validated via SDP solving. If repair and expansion fail, we default to the full half-Newton polytope, ensuring correctness. Thus, prediction errors may increase computation but cannot cause incorrect SOS certificates; Lemma 3 bounds the resulting overhead.
>
> ## Theoretical Grounding & Design Choices
>
> **Minimality**
>
> We do not claim minimality guarantees. Lemmas 6–7 provide lower bounds that are often tight on our datasets. We now state more clearly that providing tight, general upper bounds on non-minimality remains open.
>
> **Permutation scoring and over-extension**
>
> We agree $L \leq 8$ is a heuristic. For clarity, we added a concrete example from in Table 1's sparse case, where a true minimal basis has 61 monomials. Using $L=4$ permutations, the union contains 78 monomials; $L=8$ or $L=16$ both contain 82. The full Newton polytope is much larger (242 monomials). This plateau shows that extra permutations quickly stop adding new monomials.
>
>
> # Questions
>
> 	1.	Evaluation on real-world SOS problems?
> We now include a van der Pol example for real-world verification, but emphasize that the lack of suitable benchmarks remains a key obstacle.
>
> 	2.	Solver instability and heuristics?
> MOSEK encounters numerical issues on the majority of instances generated with low-rank matrices, while SCS typically succeeds on these cases. For all other structures, MOSEK remains reliable, and we recommend using it as a default with SCS as a fallback (or in parallel when resources allow).
>
> 	3.	Failure patterns for large expansion cases?
> Expansions beyond two were required only in the most challenging OOD cases. The rare, worst-case scenario arises when an essential monomial is missed across all considered permutations—prompting a fallback to the full Newton polytope to guarantee correctness. While, in principle, one could assess the similarity of a given polynomial to the training distribution to anticipate such cases, we have not yet investigated this direction.
>
> 	4.	Minimality guarantees and certification?
> We avoid claiming formal minimality guarantees. Lemmas 6 and 7 provide tight lower bounds in most settings, but bounding the gap between the learned basis and any minimal basis—especially under repair—is challenging and remains an open problem.
>
> 	5.	Efficiency on non-SOS polynomials and early termination?
> If a non-SOS case yields a strongly infeasible SDP, early termination is possible; similar behavior occurs when the proposed basis is too small. We clarify that such infeasibility-based termination does not compromise correctness, as it is conditioned on solver certificates. In Table 8, we observe that as degree and difficulty increase, the computational overhead relative to the Newton polytope drops to about $1.05$, indicating minimal overhead.
>
> 	6.	Scaling beyond 100 variables?
> We did not attempt scaling beyond 100 variables in our experiments. The current Transformer architecture can handle polynomials with approximately $4000$ terms, but at such scales the SDP solver is likely to become the main bottleneck.
>
> 	7.	Theoretical refinements to expansion scheduling or scoring?
> We used a geometric schedule for analytical simplicity and clear overhead bounds. More flexible, learned schedule via ERM are a highly interesting idea; we note this in the updated Theorem 8 discussion.
>
>
> ---
>
> We thank the reviewer for their helpful feedback, which has led to significant clarifications and improvements in the paper.

---

### Official Review · Reviewer_rzN3 · 2025-11-01

**Soundness:** 3
**Presentation:** 1
**Contribution:** 2
**Rating:** 2
**Confidence:** 2

**Summary:**

This paper addresses the challenge of certifying the non-negativity of polynomials efficiently being constructing a minimal basis based on the output of a trained transformer model and subsequently correcting the prediction to form a valid basis. The paper presents a comprehensive set of experiments to demonstrate their claims of drastic speedups for SOS programming.

**Strengths:**

**Well-Motivated and Interesting Problem**:
The problem of identifying a compact basis is, in the reviewer’s opinion, interesting, and this paper proposes a novel learning-based algorithm to address this challenge.

**Comprehensive Experimental Evaluation**:
The paper presents a range of experiments that provide strong empirical evidence supporting its claims.

**Strong Experimental Results**:
The proposed solution consistently demonstrates substantial speedups over baseline methods.

The authors also provide readable and well-documented code in the supplementary material, which aids reproducibility (although I have not attempted to reproduce the results).

**Weaknesses:**

**No Insight into the necessity of transformers**:
It is not clear why transformers, in particular, are essential to the proposed method. As presented, the approach appears applicable to any sequence-to-sequence model with a similar encoding scheme. Furthermore, the paper does not provide insights from the learned model itself that could help explain how or why the trained models can predict compact bases efficiently.

**Several missing discussions/references**:
This work does not adequately address the rich body of related work on basis selection in Polynomial Optimization. In particular, the authors should make note of and should consider baselining against the following works:

[1]. Zheng et al - Chordal decomposition in operator-splitting methods for sparse semidefinite programs, 2023

[2] Mason and Papachristodoulou - Chordal sparsity, decomposing SDPs and the Lyapunov equation, 2014

[3] Newton and Papachristodoulou - Sparse polynomial optimisation for neural network verification, 2023

[4] Ahmadi and Hall - Sum of squares basis pursuit with linear and second order cone programming, 2015

[5] Miller et al, Decomposed structured subsets for semidefinite and sum-of-squares optimization, 2022

**Lack of experiments on real-world datasets**:

While the basis pursuit for SOS verification is promising, this work suffers from a lack of applications to problems of interest that utilize polynomial optimization. Moreover, there is a lack of baselining against methods that don’t use basis search to improve efficiency - for instance, how does the end-to-end performance of the proposed transformer-based approach compare with AnySOS [6]? Can the proposed approach be used for real-time data-driven control that uses SOS programming, such as [7],[8]?

[6] Driggs and Fawzi. AnySOS: An anytime algorithm for sos programming, 2019

[7] Dai and Sznaier. A semi-algebraic optimization approach to data-driven control of continuous-time nonlinear systems, 2021

[8] Strasser and Berberich. Data-driven control of nonlinear systems: Beyond polynomial dynamics. 2021

**Questions:**

1. Since basis repair is being performed, how would the method perform when initialized with a random starting basis, as opposed to predicting them from a learnt model?

2. How does the performance of the method change across different checkpoints during training?

3. From line 693, it appears that only one epoch of training is required, suggesting very fast convergence. Could the authors provide insights into why this is the case?

4. Why would other sequence-to-sequence models, such as RNNs, LSTMs, or SSMs, not be suitable for this task?

5. Several lemmas (Lemmas 4, 5, and 6) and a theorem (Theorem 8) are presented in the appendix without reference in the main text. Could the authors include a discussion in the main body explaining the significance of these results and how they contribute to the development of the method?

6. In lines 262-263, the authors state: ‘’we exploit the fact that our Transformer is not equivariant with respect to variable orderings.’’ Could the authors provide a brief justification for this claim?

7. The paper claims that ‘’Despite being trained only on degree 12 polynomials, our model can successfully predict bases for degree 20 polynomials by leveraging familiar substructures—though it has not seen complete degree 20 monomials.’’ It remains unclear whether this behavior would extend to settings with a much larger number of variables (say 1000s).

---
Some minor comments/nitpicks that did not affect the decision:

Line 156: Specify that $Q$ is a *symmetric* matrix; this would improve clarity and rigor.

Figure 3 is presented after Figure 4; consider reordering for consistency.

**Looking forward to the discussion period to clarify these questions and strengthen this work.**

---

> ### Author Response · Authors · 2025-11-21
> **Clarifications on Transformer Motivation, Related Work, and Experimental Additions**
>
> We thank the reviewer for their valuable feedback, which helped us clarify and improve the manuscript.
> # Necessity of Transformers
> We clarified why we use Transformers, mainly because:
> 1. **Capturing global monomial interactions.**
>    Compact SOS basis selection requires modeling global monomial interactions, which Transformers efficiently capture via self-attention and long-range dependencies [9–11].
>
>    [9] Vaswani et al. Attention Is All You Need. 2017
>    [10] Tang. Why Self-Attention? A Targeted Evaluation of Neural Machine Translation Architectures. 2018
>    [11] Cheng. Simple Mathematical Word Problems Solving with Deep Learning. 2019
> 2. **Standard in symbolic and algebraic tasks.**
>    Recent work in symbolic math and polynomial reasoning consistently uses Transformers [12–14].
>
>    [12] Lample & Charton. Deep Learning for Symbolic Mathematics. 2020
>    [13] Alfarano et al. Global Lyapunov functions: a long-standing open problem in mathematics, with symbolic transformers. 2024
>    [14] Kera et al. Learning to compute Gröbner bases. 2024
>
> We have added a brief qualitative analysis: the Transformer learns intuitive patterns (e.g., including $x^2$ when $x^4$ appears) in simple cases, and finds compact bases even in more complex settings (Table 2, degree-40), outperforming naïve degree-based heuristics. However, a full mechanistic explanation remains challenging, as with most Transformer models.
> # References
> Thank you for highlighting sparsity-based SOS methods; we have expanded the related-work section accordingly.
>
> Our approach is most similar to **monomial basis reduction techniques**—such as Newton polytope pruning and diagonal-inconsistency checks—that reduce the monomial vector $z(x)$ prior to SDP construction [15].
>
> [15] Löfberg. Pre- and Post-Processing Sum-of-Squares Programs in Practice. 2009
>
> The reviewer’s highlighted methods—**chordal decomposition** [1–3] and **DSOS–SDSOS approaches** [4–5]—tackle scalability later by assuming a fixed monomial basis and leveraging matrix or correlative sparsity. Even in “basis pursuit” [4], the “basis” refers to Gram matrix structure, not reducing the monomial vector.
>
> In short, our method is **upstream** (proposes a compact basis), while decomposition and structured-subset techniques are **downstream** (improve SDP scalability). The approaches are **complementary**.
>
> # Real-world benchmarks
> We thank the reviewer for highlighting AnySOS and application-oriented work.
>
> AnySOS [6] accelerates SOS programming given a fixed SOS formulation. Our approach is **orthogonal**: we reduce the SOS problem beforehand by learning a smaller monomial basis and Gram representation.
>
> Similarly, reducing SDP size aids application papers [7, 8] that solve SOS programs, especially for real-time or data-driven control.
> Full integration is beyond scope, but we note this potential.
>
> We now include a real-world Van der Pol example, but note that standard benchmarks of sparse, fixed-coefficient SOS polynomials are lacking.
>
> # Questions
>
> **1. Random starting basis.**
> Using a random initial basis, success falls below 5% in all cases, showing the repair mechanism needs a strong initial predictor.
>
> **2. Checkpoints.**
> For easy cases, validation loss plateaus early with little improvement at later checkpoints. For harder cases, loss continues to decrease, and later checkpoints yield better recall.
>
> **3. Why only one epoch?**
> Training data _generation_ is extremely cheap. Rather than reusing a finite dataset for multiple epochs, we stream fresh examples continuously.
>
> **4. Other sequence-to-sequence models.**
> Other sequence models (RNNs, LSTMs, SSMs) could be used, but our inputs require modeling complex global dependencies. Transformers, with self-attention are better matched in this regimen [12–14].
>
> **5. Lemmas 4–6 and Theorem 8.**
> These results are now summarized and properly referenced. Lemmas 6–7 give lower bounds to support near-minimality of our predicted bases; Theorem 8 justifies the repair and extension schedule.
>
> **6. Non-equivariance.**
> Our architecture is not permutation-equivariant: variable indices are explicitly encoded, and no structural constraints enforce equivariance. Thus, for a polynomial $p$ and its permuted version $\tilde{p}$, the model usually predicts slightly different bases $B$ and $\tilde{B}$. We use this variability as a confidence signal—monomials present in both are likely confident picks, while differences suggest candidates for extension. Our extension mechanism adds monomials from $\tilde{B}$ missing in $B$; often, one such expansion suffices. With strict equivariance, $B = \tilde{B}$ always, so this signal would be lost.
>
> **7. Generalization.**
> We now state claims more cautiously. The model generalizes from degree-12 to 20 (with repair), but without repair, performance drops on harder shifts (Fig. 5(ii)). We do **not** claim scalability to thousands of variables—our results are for moderate sizes and degrees.

---

> > ### Comment · Reviewer_rzN3 · 2025-11-24
> > **First Response**
> >
> > I thank the authors for their response.
> > I agree with reviewer JFgt. It would be helpful if the authors could update the OpenReview submission according to the author guide (https://iclr.cc/Conferences/2026/AuthorGuide). A revised draft would add necessary context for several of the rebuttal responses. However, here are some immediate responses to some of the clarifications.
> >
> > ## On the necessity of transformers
> >
> > The work claims "Learning-augmented SOS Programming" (lines 102-104) as a key contribution, but reduces this to "transformer-based basis prediction" (line 175). I thank the authors for references [12-14]. However, these papers [12,13] design transformers for general mathematical representations using trees, and [14] uses a separate hybrid embedding scheme. However, these works do not exploit the special structure of polynomials. To strengthen this work, it would be helpful to demonstrate that transformers are uniquely suited to solve this problem through a simple experiment. It would be beneficial to incorporate the arguments presented in the rebuttal into the discussion in the paper as well. For example, SSMs are also known to capture long-range dependencies [16].
> >
> > [16] Mamba: Linear-Time Sequence Modeling with Selective State Spaces
> >
> > ## Regarding References
> >
> > I thank the authors for clarifying the context of the paper. It would be helpful to review these edits in the updated manuscript.
> > A discussion on the connection between the Gram matrix structure and the monomial basis would help to connect this work to classical techniques in SOS programming and broaden the paper's audience.
> >
> > ## On real-world examples
> >
> > Seeing additional details in the revised manuscript would be necessary to evaluate this result.
> >
> > ## Questions
> >
> > I thank the authors for their responses. However, the answers are very brief, and I believe further clarification is required.
> >
> > 1. Details of this experiment in the draft would be helpful. Although the success rate reduces, how would this impact the overall speedup of the monomial generation? How much cost, in terms of time, do you pay in the repair mechanism compared to the proposed technique?
> >
> > 2. An expanded discussion of this topic in the appendix, accompanied by graphs to illustrate these results, would significantly enhance the manuscript.
> >
> > 3. Please include your response to reviewer JFGt regarding the split of training time and dataset generation in the manuscript. It would help quantify the cost of the algorithm.
> >
> > 4. Refer to the previous discussion "On the necessity of transformers"
> >
> > 5. This would significantly enhance the paper. It would be helpful to review this discussion in a revised manuscript.
> >
> > 6-7. I thank the authors for clarifying the scope of the paper.
> >
> > I thank the authors for their rebuttal. I request that the reviewers update the manuscript with changes highlighted in blue.
> > The scope of the paper has been significantly clarified in the rebuttal.
> > However, I will wait to update my decision based on the edits in the manuscript.

---

> > > ### Author Response · Authors · 2025-11-26
> > > **Clarifications**
> > >
> > > # Necessity of Transformer
> > >
> > > We again thank the reviewer for their thoughtful feedback. Let us please clarify that we do not claim that the Transformer architecture is the only architecture that can be used towards our goal. To address your concern, we now explicitly state in the main text that other sequence-to-sequence models are, in principle, equally applicable. To support this point empirically, we added an ablation study in Appendix E.9 comparing the Transformer with Mamba and an LSTM baseline. In our setup, the Transformer converged fastest, Mamba performed competitively, and the LSTM converged more slowly but still reached reasonable accuracy, please see the revision for exact results. We have incorporated these observations directly into the main discussion, as suggested.
> > >
> > > # On the random initial basis setup
> > >
> > > We have added experiments that initialize from both a constant and a random basis in Appendix E.8. Consistent with the reviewer’s concern, we observe a very low success rate. Moreover, this setting introduces substantial overhead compared to the Newton baseline—ranging from 1.96× to 3.93×—and the method times out for the larger configuration. These results lead us to conclude that random bases are insufficient for our approach, even with a subsequent repair step. The revised manuscript includes a clearer explanation of this finding, along with the results in Table 29.

---

### Author Response · Authors · 2025-11-26
**Overview**

**We sincerely thank all reviewers for their thoughtful and constructive feedback. We have uploaded the revised manuscript with all changes highlighted in blue. Below, we briefly list the major changes made to the manuscript.**


### 1. Are Transformers the only suitable architecture for our task?

As noted in Remark 1 in the main text, our pipeline is not specific to Transformers—it only requires a sequence-to-sequence model. *Appendix E.9* compares Transformer, LSTM, and Mamba under identical training conditions (5 hours, comparable parameter counts). Transformers achieve the lowest test error (0.038) and fastest convergence; Mamba is close (0.041); LSTM is slower and less accurate (0.067). While Transformers perform best, other seq2seq models are competitive.

---

### 2. Random/Uninformative Initialization

*Appendix E.8* tests whether repair alone suffices without a learned initializer. We evaluated two baselines: (i) constant basis $\{1\}$, and (ii) uniformly random basis from the half Newton polytope.

Greedy repair *cannot* compensate for an uninformative starting point. A meaningful learned initializer is essential.

---

### 3. Training Cost Amortization

*Appendix E.12* provides an amortization analysis across four problem scales.

For the challenging regimes where classical SOS struggles most, training pays off after **6–8 instances**—precisely where our approach provides the most value.

---

### 4. Real-World Applicability: Van der Pol Oscillator

*Appendix E.11* presents a small control-theory-based case study. For a degree-4 Lyapunov candidate from the Van der Pol oscillator, SumOfSquares.jl (using the Newton polytope) produces a 4-monomial basis, whereas our method recovers the minimal 3-monomial basis. This is a single case, but it shows the approach can generalize to real dynamical systems. The lack of established SOS benchmarks in this domain remains a challenge.

---

### 5. Theoretical Integration and Clarity

We reorganized the presentation to make the theory more prominent:

- **Lemmas 4–6** now have explicit explanations of how they inform repair and provide near-minimality guarantees for training data.
- **Theorem 8** (geometric expansion) is contextualized with clear statements about worst-case overhead bounds ($O(k)$ factor) and the ERM-based approach for selecting $\rho$.


---

### 6. Permutation Scoring and Expansion Schedules
We added a concrete example: for a sparse instance with a 61-monomial minimal basis, the union from $L{=}4$ permutations yields 78 candidates; $L{=}8$ or $L{=}16$ both plateau at 82—far smaller than the 242-monomial Newton polytope. Moderate $L$ rapidly saturates the candidate set.

The geometric expansion schedule is chosen for analytical tractability, but the framework accommodates learned or adaptive schedules (Theorem 8, Remark following).

---

### 7. Expanded Related Work

The related-work section now distinguishes our **upstream** basis-reduction approach from **downstream** methods:

- **Chordal decomposition, term sparsity (TSSOS), structured subsets:** improve SDP scalability given a fixed basis
- **DSOS/SDSOS:** replace SDP with LP/SOCP at the cost of tightness
- **AnySOS:** accelerates SDP solving for fixed formulations

Our learned basis selection is complementary—a compact upstream basis reduces problem size before these downstream techniques apply.

---

### Additional Revisions

- **Checkpoint analysis** (*Appendix E.11*): Performance evolution across training, with error dropping by >10× from 10K to 1M samples.
- **Qualitative model behavior** (*Appendix E.14*): The Transformer learns intuitive algebraic patterns (e.g., including $x_i^2$ when $x_i^4$ appears).
- **Other:** We now explicitly state that we do *not* claim scalability to thousands of variables, and that formal minimality guarantees remain open.

---

We believe these revisions substantially address the reviewers' concerns regarding experimental validation, theoretical grounding, and practical applicability. We are grateful for the constructive dialogue that allowed us to improve our manuscript.

---

### Meta-Review · Area_Chair_sYrj · 2026-01-07

**Summary:**

The paper proposed a learning-augmented algorithm for Sum-of-Squares (SOS) certification that leverages a Transformer to predict compact monomial bases, thereby drastically reducing the dimensionality and computational cost of the resulting Semidefinite Programs (SDP) .

Overall, the topic is unique and interesting and the proposed method shows impressive performance. As the reviewers said, the approach creatively integrates deep learning into a classical symbolic optimization bottleneck, demonstrating impressive speedups and scalability compared to state-of-the-art solvers .

All the reviewers provide detailed review, and the authors made substantial effort to improve the draft, particularly by incorporating a real-world Van der Pol oscillator case study , quantifying training cost amortization , and rigorously ablating the learned component against random baselines. These revisions successfully addressed concerns regarding synthetic data reliance and established the method's practical utility.

**Reviewer Concerns:**

### [rzN3]

References: The authors' rebuttal incorrectly dismisses references [4] (Ahmadi & Hall) and [5] (Miller et al.) as "downstream" methods that operate on a "fixed monomial basis." This characterization is factually inaccurate. Both works explicitly integrate Basis Pursuit—an iterative optimization strategy that dynamically updates the basis. They are not merely static solvers but are direct competitors in the "upstream" task of identifying efficient basis representations. The distinction is that [4] and [5] use iterative convex optimization to adapt the basis, whereas the current work uses a learned neural heuristic. The final camera-ready version must correct this distinction.

Other concerns are addressed.

### [wmbZ]

All concerns are addressed.

### [JFgt]

All concerns are addressed.

**Reviewer Scores:**

The AC anticipates anticipates the following outcomes:

Reviewer rzN3 is likely to raise their score from 2 to 4 (or potentially 6), as most of their concerns have been effectively addressed.

Reviewers wmbZ and JFgt currently rate the paper a 6.

Given JFgt’s active engagement and openness to re-evaluation, JFgt is likely to increase their score to 8 (or at minimum, maintain the 6).

---

### Decision · Program_Chairs · 2026-01-26

Accept (Poster)